# Weaves, Wires, and Morphisms: Formalizing and Implementing the Algebra of Deep Learning

## Abstract

Despite deep learning models running well-defined mathematical functions, we lack a formal mathematical framework for describing model architectures. Ad-hoc notation, diagrams, and pseudocode poorly handle nonlinear broadcasting and the relationship between individual components and composed models. This paper introduces a categorical framework for deep learning models that formalizes broadcasting through the novel axis-stride and array-broadcasted categories. This allows the mathematical function underlying architectures to be precisely expressed and manipulated in a compositional manner. These mathematical definitions are translated into human manageable diagrams and machine manageable data structures. We provide a mirrored implementation in Python and TypeScript to show the universal aspect of our framework, along with features including algebraic construction, graph conversion, PyTorch compilation and diagram rendering. This lays the foundation for a systematic, formal approach to deep learning model design and analysis.

## 1 Introduction

Deep learning models implement precisely defined mathematical computations, yet the way we describe model architectures remains surprisingly informal. In practice, architectures are communicated through a mixture of tensor notation, framework-specific code, and hand-drawn diagrams. These representations are useful locally, but they do not provide a single formal object on which one can systematically reason. As a result, properties that should in principle be derivable from a model's mathematical definition, such as equivalent formulations, efficient low-level implementations, performance models, or realizations in multiple software frameworks, are often discovered through manual derivation and engineering intuition rather than obtained procedurally. A more explicit mathematical language for architectures would make these relationships easier to analyze, compare, and eventually automate.

Deep learning models implement mathematical mappings from model inputs $x$ to outputs $\hat{y}$. These mappings are typically organized as a series of residual connections (He et al., 2015), each implementing $x_{i+1} = f_i(x_i) + x_i$, where each layer $f_i$ is composed of various linear layers typically in an attention (Vaswani et al., 2017) or feed-forward arrangement. This mapping–which we will call the model architecture–is precisely defined from these formulas. The mysterious "black-box" behavior we associate with neural networks is an emergent property and can be separated from the study of architectures in isolation.

Understanding architectures is critical to many of the engineering aspects of deep learning models. A given architecture can be implemented in various ways, with multiple computational sequences implementing the same underlying function. For example, *FlashAttention* (Dao et al., 2022) implements attention in a parallelized, hardware-aware approach which yields the same mathematical result at far greater efficiency and throughput. The power of deep learning models relies on the ability to deploy an immense quantity of compute towards a task, which restricts the space of reasonable architectures to mathematical functions which can be computed on parallelized hardware. The need for parallelized hardware stems from fundamental physical limitations on the clock speed and thermodynamic characteristics of individual processors, meaning only parallized, GPU hardware can deliver the needed compute. Designing architectures for parallelized GPU hardware ushered in modern deep learning with *AlexNet* (Krizhevsky et al., 2017), allowed

for LLMs to compute results in parallelized batches with transformers as opposed to sequential recurrent neural networks (Vaswani et al., 2017), while newer models such as *DeepSeek-V3* (DeepSeek-AI, 2025) have achieved remarkable efficiency gains by better co-designing software with hardware. As parallelization is the key aspect of deep learning architectures that enables the necessary quantities of compute to be deployed, parallelization is a critical aspect of understanding architectures.

Understanding architectures and their parallelization properties is therefore a key aspect of innovation in deep learning, however, these innovations have so far proven elusive, indicating that the understanding of architectures is a gap For instance, *FlashAttention* (Dao et al., 2022) was first released five years after attention mechanisms (Vaswani et al., 2017), and required further refinement and years-long lags to catch up with the latest hardware (Dao, 2024; Shah et al., 2024; Zadouri et al., 2026) *despite* the underlying mathematical architecture (attention) being unchanged. *DeepSeek-V3*'s innovations delivered near state-of-the-art results at a fraction of the cost of models from the largest labs, emphasizing the difficulty of architecture-hardware co-design.

In our view, the key missing aspect of mathematically understanding deep learning architectures in a manner useful for the engineering challenge of co-designing parallelized algorithms is understanding broadcasting. Broadcasting refers to how an operation is extended over additional axes. For example, SoftMax is a polymorphic operation defined by $\mathbb{R}^x \to \mathbb{R}^x$. When this is taken to be a row-wise operation applied to each of $n$-rows of a tensor, it will then have the form $\mathbb{R}^{nx} \to \mathbb{R}^{nx}$. This fundamentally changes the mathematical character of the operation, and we see immediately where confusion can arise. For a useful mathematical understanding to be developed, we need to know whether $x$ or $n$ were the broadcasted or target axes. For another example, a dot product $\mathbb{R}^b \times \mathbb{R}^b \to \mathbb{R}$ (which we will refer to as contraction) performed in parallel over $a$-rows for the first input and $c$-columns for the second results in matrix multiplication of the form $\mathbb{R}^{ab} \times \mathbb{R}^{bc} \to \mathbb{R}^{ac}$. *Batched* matrix multiplication over an additional $n$-sized axis then has a form $\mathbb{R}^{nab} \times \mathbb{R}^{nbc} \to \mathbb{R}^{nac}$. Convolved matrix multiplication again has its own complex shape, taking $\mathbb{R}^{xc_{in}} \times \mathbb{R}^{wc_{in}c_{out}} \to \mathbb{R}^{(x+w-1)c_{out}}$. We see that broadcasting can take various shapes and forms, and any hope of mathematically understanding this behavior and therefore the parallel character of deep learning models requires clearly distinguishing and defining these forms for mathematical manipulation.

What are the benefits of understanding this behavior? At the most basic level, clearly defining broadcasting is required to simply know what we mean when we say we have an $\mathbb{R}^{nx} \to \mathbb{R}^{nx}$ SoftMax operation. Without the broadcasting information, the expression may be implicitly known but is technically ambiguous and therefore impervious to robust mathematical analysis. Broadcasting is key to understanding how mathematical architectures map to parallelized hardware, and kernels (tiled algorithms which compute a parallelized result) are defined with respect to the broadcasted axes. Performance models can be derived from these kernels, and therefore clearly defined broadcasting allows performance modeling to be a matter of applying consistent mathematical rules. Additionally, in practice deep learning models are bandwidth and memory bound (Gholami et al.). Tricks which reduce bandwidth and memory constraints are key to innovations such as *FlashAttention* and *DeepSeek-V3*. As is shown in *FlashAttention on a Napkin* (Abbott & Zardini, 2025), clearly showing broadcasting allows FlashAttention and its performance model to be procedurally derived. The techniques in *FlashAttention on a Napkin* have subsequently been used to develop low-level algorithms for attention variants (Abbott et al., 2025), showing the utility of such an approach. This is an immense gain over the manual derivation typically required to derive such algorithms and opens up the door to automatically derived low-level kernels and performance models, an achievement which escapes the compilation tools of packages such as PyTorch. Fundamentally, as parallelized computing is *the* critical property of architectures which overcomes the physical constraints of single core processing and therefore enables deep learning, broadcasting is *the* critical mathematical property of deep learning architectures as opposed to other subjects of mathematical study, similar to how symmetry is the critical property of many physical systems.

Standard methods of describing broadcasting are haphazard and fall short of the task of expressing deep learning models as they appear in practice. Standard deep learning notation borrows from linear algebra (Goodfellow et al., 2016). This works somewhat for simple matrix multiplication, can be elegantly extended with Einstein notation (Rogozhnikov, 2022), but completely fails for non-linear operations such as SoftMax. Named tensor notation or explicit re-expressions (Chiang et al., 2023; Phuong & Hutter, 2022)

provide additional clarity but provide no additional tools of mathematical analysis. Neural circuit diagrams (Abbott, 2024) fully describe deep learning models and supply useful blueprinting diagrams. They are the basis of the successful *FlashAttention on a Napkin* approach (Abbott & Zardini, 2025), showing their utility for mathematical analysis and deriving efficient algorithms, and have indeed been used to construct new algorithms Abbott et al. (2025). However, they so far lack a formal mathematical representation outside of diagrams which can be used for automated computational analysis.

Developing such a formal mathematical representation coupled with an appropriate code implementation requires the use of category theory. Category theory is the mathematics of composition and abstraction (Censi et al., 2024). Just as group theory's theoretical tools become pertinent for physical systems characterized by symmetries, category theory's compositional and layered abstraction tools become pertinent for composed deep learning systems that need to be analyzed at multiple levels of abstraction (mathematical architecture, high-level algorithm, low-level code). This is especially relevant to tackling understanding the parallel nature of deep learning architectures as parallelization is itself a composition property–sequentially composed row-wise operations can be interpreted as a single, composed row-wise operation. Category theory provides a language for composed systems by identifying *morphisms*, fundamental compositional units, which are anchored by *objects*, that restrict the domain and codomain of allowed compositions. This is typically extended by *products*, which allow for parallel constructs. When composition and products are closed, we form a category, and this structure covers various mathematical constructs such as functions anchored by sets, the category **Set**, linear maps anchored by linear spaces, **Vect**, or Markov kernels anchored by probability spaces, **Stoch** (Fritz et al., 2023). Systems defined by composition and products follow standard rules, allowing access to a "library" of robust, standard mathematical transformations between various lenses of analysis such as converting an expression into formal diagrams or hypergraphs (Piedeleu & Zanasi, 2025). Category theory has been used to describe aspects of deep learning models, with backpropagated algorithms and data types being described by **Para** (Fong et al., 2019; Cruttwell et al., 2024; 2022; Gavranović et al., 2024; Shiebler et al., 2021), model symmetries and equivariances being approached with geometric deep learning (Veličković et al., 2018; Bronstein et al., 2021), and even to describe the algebra of CuTe layouts for low-level execution (Research, 2025). The co-design of complex systems, a necessary goal of architectural analysis to have engineering utility, can be viewed through the resource/functionality composition of the **DP** category of design problems (Zardini, 2023). A categorical approach, then, has the utility of allowing for an integrated web of tools, **Vect** allows for linear components of models to be rearranged, **Stoch** provides a lens for the probabilistic aspects of dropout or quantization, **Para** allows for backpropagation to be natively built from a category-theoeretic lens, and **DP** would allow for engineered co-design of deep learning software and hardware.

By using category theory to define broadcasting coupled with a code implementation, we aim to lay the foundation for deep learning architectures to be clearly defined in a manner that opens up powerful, robust tools for automated analysis. The scope of this paper is to (1) establish how mathematical definitions can be converted to reliable representations across written expressions, diagrams, and code (Section 2.2.1) (2) establish how a generic product category can be fit within this framework, granting access to categorical compositional construction (Section 3), (3) describe the broadcasted-array category **Br** which is able to express parallelized, broadcasted operations and finally (Section 4.2) (4) show how this framework can be used as the basis for an implementation with an integrated suite of features, including describing the full architecture of *DeepSeek-V3*, a major model used in practice including attention, various datatypes, and mixtures of experts (Section 6). This supplies both a concrete mathematical description of deep learning models, as used in practice with critical broadcasting information clearly outlined, coupled with a modular extendable implementation to be integrated into future tools.

## 2 Encoding Mathematics

A formal framework for deep learning is only useful if the resulting objects can be manipulated by both humans and machines. In practice, we want to move between at least three views of the same object: a mathematical definition, a human-facing representation such as notation or a diagram, and a machine-facing representation such as code. This section introduces a general framework for doing so. The key idea is to separate the mathematical objects we wish to describe from the concrete terms used to represent them.

The **constructed term framework** we introduce here as many desirable properties. It ensures that mathematical definitions correspond to representations, whether that be in formal written expressions, diagrams, or code. The generated expressions allow for open variables, enabling algebra to be performed and expressions to represent a polymorphic class of expressions with variable axis sizes that are open to configuration. This means that diagrams, code, and mathematics represent deep learning architectures as they are used in practice. Some axes may be non-static variable sizes depending on the input, while others may be configured to a specific size during deployment.

## 2.1 Nominal Mathematical Entities and Real Terms

We begin with a set $\Gamma$ of nominal mathematical entities representing the platonic mathematical expressions we wish to implement. These entities are equipped with a family of basic structure maps, which we call **core properties**. For each $k \in K$, a core property has the form $\pi_k : \Gamma_{k,i} \to \Gamma_{k,f}$, where $\Gamma_{k,i}, \Gamma_{k,f} \subseteq \Gamma$ specify the domain on which the property is defined and the codomain in which it lands. We also assume that finite products of entities are again entities, so that $\prod_{i \in I} \Gamma \subseteq \Gamma$. This lets us express multi-input structure without leaving the ambient space of entities.

**Definition 1** (Mathematical Entities). *A system of mathematical entities consists of:*

- *A set of **entities** $\Gamma$. We include finite products of entities, so that $\prod_{i \in I} \Gamma \subseteq \Gamma$.*

- *A $K$-family of **core properties** $\pi_k : \Gamma_{k,i} \to \Gamma_{k,f}$ so that $\Gamma_{k,i} = \operatorname{dom} \pi_k$ and $\Gamma_{k,f} = \operatorname{img} \pi_k$*

A **constructed term system** is a representation layer for $\Gamma$, providing real terms and expressions we can implement, visualize and manipulate while guaranteeing correspondence to an underlying mathematical system. Its terms may be symbolic expressions, diagrams, or code objects, but in every case they must faithfully represent the same underlying entities. Formally, we introduce a set of terms $G$ together with an interpretation function $V_G : G \to \Gamma$. The map $V_G$ tells us which mathematical entity a term denotes. As with entities, we include finite products of terms and interpret them componentwise:

$$\left( \prod_{i \in I} g_i \right) \fatsemi V_G = \prod_{i \in I} \left( g_i \fatsemi V_G \right).$$

For each core property $\pi_k$, the term system should provide an internal counterpart

$$G_{k,i} := \operatorname{preimg} V_G(\Gamma_{k,i}), \qquad G_{k,f} := \operatorname{img} p_k \subseteq \Gamma_{k,f} \qquad p_k : G_{k,i} \to G_{k,f}, \qquad p_k \fatsemi V_G = V_G \fatsemi \pi_k$$

so that evaluating inside the term system agrees with evaluating in the mathematical space after interpretation. This is the basic soundness condition of the framework.

**Definition 2** (Constructed Term System). *A constructed term system consists of:*

- *A set of **terms** $G$, again closed under finite products: $\prod_{i \in I} G \subseteq G$.*

- *An **interpretation function** $V_G : G \to \Gamma$. On elements $g \in G$, we may write $g \fatsemi V_G = [\![g]\!]$.*

- *For each $k \in K$, an internal **core property** $p_k : G_{k,i} \to G_{k,f}$ defined over $G_{k,i} := \operatorname{preimg} V_G(\Gamma_{k,i})$ and $G_{k,f} := \operatorname{img} p_k \subseteq \operatorname{preimg} \Gamma_{k,f}$, satisfying **internal evaluation**, $p_k \fatsemi V_G = V_G \fatsemi \pi_k$.*

## 2.2 Implementing Core Properties

Operationally, terms natively carry information within subexpressions. In a visual, human readable form these are notated, while in code these are fields inside data structures, and in either case we have access to native extraction functions. We use two complementary methods to implement core properties which ensure extendability and soundness as described above. In one case, we use **construction rules**, so that a term remembers the inputs from which it was built, contravariantly storing $G_{k,i}$ in $G_{k,f}$ (reverse of $p_k$). In the other, we use **root terms**, storing the data needed to expose certain properties directly, covariantly containing $G_{k,f}$ in $G_{k,i}$ (forward of $p_k$).

### 2.2.1 Construction Rules

We start with the contravariant case. We choose a subset of core properties $C \subseteq K$ to be implemented as **construction rules**, $p_c = T_c : G_{c,i} \to G_{c,f}$ so that we can extract builder terms via a native extraction function $\hat{T}_c : G_{c,f} \to G_{c,i}$. In other words, $T_c$ operationally acts as a wrapper that serves the role of indicating the construction rules used and remembering the inputs provided, think of how the expression "$3+2$" indicates a construction rule ($+$) along with builder terms (2 and 3). Then, the set of terms $G_{c,f}$ corresponds to data types or expressions generated by this rule i.e. those that contain a "$+$" symbol. For soundness, we require that $\hat{T}_c \,\mathbin{\fatsemi}\, T_c = \mathrm{Id}_{G_{c,f}}$, meaning that reconstruction yields the same expression, and that $V_{G_{c,f}} : G_{c,f} \to \Gamma_{c,f}$ is defined by $V_{G_{c,f}} = \hat{T}_c \,\mathbin{\fatsemi}\, \pi_k$, ensuring the construction rule nominally corresponds to the underlying core property.

**Definition 3** (Construction Rules). *For a chosen subset $C \subseteq K$ of core properties, we implement $p_c$ for $c \in C$ as a construction rule $T_c : G_{c,i} \to G_{c,f}$ so that:*

- *We have a native recovery function $\hat{T}_c : G_{c,f} \to G_{c,i}$ so that $\hat{T}_c \mathbin{\fatsemi} T_c = Id_{G_{c,f}}$ (i.e., for every $g \in \mathrm{img}\, T_c$, we have $g \mathbin{\fatsemi} \hat{T}_c \mathbin{\fatsemi} T_c = g$). This formalizes the notion that constructed term contains its builder terms.*

- *Interpretation is defined by $V_{G_{c,f}} = \hat{T}_c \mathbin{\fatsemi} V_G \mathbin{\fatsemi} \pi_c$.*

Compound construction rules are built from placing rules in parallel or composing them. For parallel rules $T_c \times T_d$, we have a recovery

$$\hat{T}_c \times \hat{T}_d : G_{c,f} \times G_{d,f} \to G_{c,i} \times G_{d,i}.$$

In the case that $G_{c,f} \subseteq G_{d,i}$, we have a composed rule $T_c \mathbin{\fatsemi} T_d$ with a recovery;

$$(T_c \mathbin{\hat{\fatsemi}} T_d) = \hat{T}_d \mathbin{\fatsemi} \hat{T}_c : G_{d,f} \to G_{c,i}.$$

In both cases, recovery and soundness holds. As compound rules are built from these elemental combinations, complex expressions built from construction rules can be decomposed and interpreted as non-constructed root terms.

In addition to providing soundness and interpretation from merely defining a wrapper, these construction rules grant as access to **axioms**. An axiom is a statement that two construction methods yield the same result. Consider associativity, where for instance we have $[\![(x+y)+z]\!] = [\![x+(y+z)]\!]$. When this occurs, we can use an axiomatically unified expression $x + y + z$ which can be interpreted to the correct result. As the reconstruction rules are not guaranteed in the forward case, $T_c \mathbin{\fatsemi} \hat{T}_c \neq \mathrm{Id}_{G_{c,i}}$ we do not need to natively maintain all the construction information, but merely that which is sufficient for $\pi_c$ to provide a correct interpretation. Axiomatic unification stems from the underlying nominal mathematical entities as, over the unified term $g_{c \cap d}$, it implies that over $\gamma_c = g_{c \cap d} \mathbin{\fatsemi} \hat{T}_c$ and $\gamma_d = g_{c \cap d} \mathbin{\fatsemi} \hat{T}_d$ we have $\gamma_c \mathbin{\fatsemi} \pi_c = \gamma_d \mathbin{\fatsemi} \pi_d$. In code implementations, construction rules rarely generate axioms. However, in diagrams we often use these principles to simplify expressions.

### 2.3 Root Terms

For non-constructed rules $\ell \in K \backslash C$, we do not use simple wrappers. Rather, we either embed $G_{\ell,f}$ in non-constructed **root terms** or define a method over applicable constructed terms. Non-constructed root terms are separated into **root types**. A root type corresponds to a subset $r \subseteq K \backslash C$ of construction rules for which only those are applicable. Terms in root types $g_r \in G_r$ are embedded with data which can derive all $\ell \in r$.

**Definition 4** (Root Terms). *For a subset $r \subseteq K \backslash C$, the root term type $r$ covers all terms with the $r$-collection of core properties which cannot be generated by a construction rule in the non-identity case i.e. when a construction rule's axioms leaves an expression unchanged. Hence, they describe a certain collection of non-constructable terms which require core properties to be natively supplied.*

$$G_r = \left\{ g \in G \mid (\forall \ell \in K \backslash C.g \mathbin{\fatsemi} V_G \in \Gamma_\ell \leftrightarrow \ell \in r) \wedge (\forall \ell \in C.g \notin G_{c,f} \vee g = g \mathbin{\fatsemi} T_c) \right\}.$$

*For the root term type $r$, we define $p_{\ell,r} : G_r \to G_{\ell,f}$ for $\ell \in r$ via native extraction.*

| Term Type | Associated Data | Constructed Property | Non-constructed Property | Interpretation |
|---|---|---|---|---|
| Product | $g_{\Pi j} = \Pi_{j\in J} g_j$ | *Wrapping* | Method $m_{\Pi j,\ell} : \Pi_{j\in J} G_j \to G_{\ell,i}$ $\pi_\ell = m_{c,\ell} \mathbin{\fatsemi} V_G$ | $(\Pi_{j\in J} g_j) \mathbin{\fatsemi} V_G = \Pi_{j\in J}(g_j \mathbin{\fatsemi} V_G)$ |
| Constructed Term | $g_{c,f} \mapsto g_{c,i}$ via $\hat{T}_c$ | *Wrapping* | Method $m_{c,\ell} : G_{c,i} \to G_{\ell,f}$ $\pi_\ell = \hat{T}_c \mathbin{\fatsemi} m_{c,\ell} \mathbin{\fatsemi} V_G$ | $g_{c,f} \mathbin{\fatsemi} V_G = g_{c,f} \mathbin{\fatsemi} \hat{T}_c \mathbin{\fatsemi} V_G \mathbin{\fatsemi} \pi_c$ |
| Root Term | $g_r \mapsto \mu_r \times \Pi_{\ell\in r} g_{\ell,f}$ *Where $m_r$ is non-core metadata.* | *Wrapping* | *Data Extraction* $p_{r,\ell} : G_r \to G_{r,\ell}$ $\pi_\ell = p_{r,\ell} \mathbin{\fatsemi} V_G$ | $g_r \mathbin{\fatsemi} V_G = \gamma_r$ $\forall \ell \in r. g_r \mathbin{\fatsemi} p_{r,\ell} \mathbin{\fatsemi} V_G = \gamma_r \mathbin{\fatsemi} \pi_\ell$ |

Table 1: Applicability and interpretation for the main term types.

Interpretation of a root term $g_r \in G_r$ results in an entity $\gamma_r \in \Gamma_r$ so that core properties correspond via $p_{\ell,r} \mathbin{\fatsemi} V_G = V_G \mathbin{\fatsemi} \pi_\ell$. As we construct the data of terms by assigning $g_{\ell,f}$ properties, we use **templates** to restrict the creation of this data only to terms which indeed correspond to an entity. Entities may differ on some non-core property which is not the focus of the representation. In this case, the root terms are equipped with **metadata** tags which can be used by some alternative representation.

For non-constructed properties $\ell \in K\backslash C$ of non-identity constructed terms $g \in G_{\ell,i}$, we must define **methods**. These are functions $m_{c,\ell} : G_{c,i} \to G_{\ell,f}$ so that the property $p_\ell$ can be derived with respect to the extracted builder terms via $\pi_\ell = \hat{T}_c \mathbin{\fatsemi} m_{c,\ell} \mathbin{\fatsemi} V_G$. These functions are non-native, and must be defined according to the specific construction rule and relevant property.

Overall, this leads to a table of applicability and interpretation over all terms (Table 2.3).

### 2.4  Placeholder Terms

Many useful expressions are only partially instantiated. For example, the expression "$x + y + z$" does not denote a concrete real numbers; instead, it denotes a construction pattern with three open inputs. In the present framework, this corresponds to a compound construction rule $T_{x+y+z} : \mathbb{R}^3 \to \mathbb{R}$ whose inputs have not yet been fixed. Such partially constructed terms are important because they preserve the degrees of freedom in an expression while still exposing its compositional structure.

In symbolic notation and diagrams, these open slots appear as free symbols. In code, they appear as terms equipped with unique identifiers (UIDs). This lets us manipulate expressions symbolically before committing to concrete inputs. For instance, imposing the substitution $y := z$ transforms the expression "$x + y + z$" into "$x + y + y$", and correspondingly transforms $T_{x+y+z} : \mathbb{R}^3 \to \mathbb{R}$ into $T_{x+y+y} : \mathbb{R}^2 \to \mathbb{R}$. Equivalently, the substitution may be viewed as a meta-level rearrangement $[0,1,1]_{(\mathbb{R})_{i\in 3}} : \mathbb{R}^2 \to \mathbb{R}^3$ (denoting the first output maps from the first input, and second and third outputs map from the second input), anticipating the rearrangement structure introduced in Section 3.

By equipping the term system with placeholder terms, we obtain a uniform account of symbolic manipulation across notation, diagrams, and code. We can scan an expression for free symbols or UID-tagged terms, generate the corresponding configuration space automatically, and impose canonical substitutions when performing algebraic simplifications. Furthermore, it means that the partially constructed terms generated by our expression with free inputs, whether in diagrams or code, represent polymorphic expressions which accomodate a range of variables.

## 3  Categories

Category theory provides a descriptive framework for managing compositional systems. Many mathematical structures follow a common pattern, with a key compositional operation accompanied by parallel products.

For instance, functions compose sets and can be put in Cartesian parallel (the category **Set**). Markov kernels propagate probability distributions across probability spaces (the category **Stoch**). Kernels independently applied over joint probability spaces then yield products. Directed graphs can be composed in sequence, should their targets and origins match, and these graphs and nodes can be put in parallel. Category theory provides the minimal descriptive tools to describe the compositional structure of these systems, and thereby yields universal mathematical properties applicable across them all.

For analyzing deep learning models, these tools become invaluable. A compositional description of a deep learning model serves as a template for various interpretations to be applied–it can be viewed as a pure mathematical function, interpreted as a probabilistic evolution, manipulated as an algebraic graph, or transformed into algorithmic instructions. As outlined in the background (see Section 1), this allows for a robust library of tools to be developed. The scope of this work is to provide the basic mathematical framework on which such tools can be based.

We begin with a basic outline of a monoidal product category equipped with rearrangements, which serves as a template for various systems that will be specific in Section 4.1.3 to describe deep learning models. This template views a system as being composed of **morphisms** each of which has a **domain** and **codomain** object. When these match, we have **composition**, an associative operation equipped with identities. **Products** place these in parallel with certain compatibility requirements. For a concrete example, the category **Set** has functions as morphisms. Each function has a domain and codomain set, which serve as objects. Composition is sequential function application. The product of objects is provided by Cartesian products, and the product of morphisms has them act on separate tuple segments. As sequential application is associative, and the parallel application of composed functions is equivalent to the sequential composition of parallel functions, **bifunctoriality** as the product compatibility requirement is well-defined.

**Definition 5** (Monoidal Product Category). *A monoidal category $\mathcal{C}$ has;*

- ***Objects.*** *A collection of objects $A \in Ob\mathcal{C}$. The product of objects $\otimes : Ob\mathcal{C} \times Ob\mathcal{C} \to Ob\mathcal{C}$ is an associative operation with a distinguished unit object $\mathbb{1} \in Ob\mathcal{C}$.*

- ***Morphisms.*** *A collection of morphisms $f \in Mo\mathcal{C}$, so that each morphism has a **domain** and **codomain** object. The collection of morphisms with domain $A$ and codomain $B$ are denoted $\mathcal{C}(A,B)$, and we write $f \in \mathcal{C}(A,B)$ as $f : A \to B$.*
    - *The composition of morphisms $\mathbin{\raise0.2ex\hbox{$\scriptstyle\circ$}} : \mathcal{C}(A,B) \times \mathcal{C}(B,C)$ is an associative operation. Each object has an identity morphism, so that $f \mathbin{\raise0.2ex\hbox{$\scriptstyle\circ$}} Id_B = f = Id_A \mathbin{\raise0.2ex\hbox{$\scriptstyle\circ$}} f$.*
    - *The product of morphisms $f : A \to B$ and $g : C \to D$ yields $f \otimes g : A \otimes C \to B \otimes D$ is an associative operation with $Id_{\mathbb{1}}$ serving as the unit.*
    - *These exhibit **bifunctoriality**, so that $(f \otimes h) \mathbin{\raise0.2ex\hbox{$\scriptstyle\circ$}} (g \otimes k) = (f \mathbin{\raise0.2ex\hbox{$\scriptstyle\circ$}} g) \otimes (h \mathbin{\raise0.2ex\hbox{$\scriptstyle\circ$}} k)$ when the compositions are well-defined.*

- ***Structural Guarantees*** *ensuring associativty of products and the behaviour of the unit object (Selinger, 2010).*

Monoidal product categories are additional equipped with **rearrangements**. These are structural morphisms corresponding to swapping (count neutral), copying (count increase), and deleting (count decrease) information. For instance, copying provides us with $[0,0]_X : X \to X \otimes X$. The tuple $[0,0]$ (the remapping) indicates where output data comes from, and can be viewed as a finite function $\mu : J \to I$. Swapping can be shown by $[1,0]_{(X,Y)} : X \otimes Y \to Y \otimes X$ and deletion by $[]_X : X \to \mathbb{1}$. Monoidal product categories differ in which rearrangements are allowed, and which rearrangements are assigned the structurally critical **naturality** property. If, for instance, count increase (copying) is natural then copying after a morphism is equivalent to copying beforehand and running it twice, so that $f \mathbin{\raise0.2ex\hbox{$\scriptstyle\circ$}} [0,0]_Y = [0,0]_X \mathbin{\raise0.2ex\hbox{$\scriptstyle\circ$}} (f \otimes f)$. By indicating which rearrangements are natural, we can yield generic structural templates. For instance, in **Set** copying is natural – copying the result of a function is equivalent to copying the input and running the function twice. In **Stoch** copying is not natural, consider that copying the result of a dice roll yields a distinct distribution from rolling two independent dice. This describes other systems as well – the computational cost of copying the result of a function is distinct from copying an input and running the function twice, and copying the

result from a single linear layer is architecturally distinct from copying an input and feeding it to two distinctly trained linear layers. In the typical case of a symmetric monoidal category, swapping (count neutral rearrangements) is allowed and natural, and yields a graph structure Piedeleu & Zanasi (2025). Rearrangements serve to embed this structural information into our template and the resulting mathematics, allowing our diagrams and code implementations to consider these characteristics.

**Definition 6** (Rearrangements). *A monoidal category is equipped with allowed remappings, finite functions $\mu : I \to J$, a subset of which are considered **natural**. Remappings are classified by their **count**, which indicates the number of overlaps between inputs. Neutral counts have each input appear once in the outputs, **increased count** allows for inputs to appear multiple times, and **decreased count** allows inputs to not be present in the outputs.*

$$\text{Count}^{=}(\mu) = \{\forall j \in J. |\{i \in I. \mu(i) = j\}| = 1\}$$
$$\text{Count}^{+}(\mu) = \{\exists j \in J. |\{i \in I. \mu(i) = j\}| > 1\}$$
$$\text{Count}^{\text{-}}(\mu) = \{\exists j \in J. |\{i \in I. \mu(i) = j\}| < 1\}$$

*Given a family of objects $(A_i)_{i \in I}$, allowed remappings generate rearrangement morphisms $[\mu]_{(A_i)_{i \in I}} : \Pi_{i \in I} A_i \to \Pi_{j \in J} A_{\mu(j)}$ so that;*

- *Identity $\mu = Id_I : I \to I$ reamappings are always allowed and natural, and generate identity morphisms, $[Id_I]_{(A_i)_{i \in I}} = Id_{\Pi_{i \in I} A_i}$*

- *The product with another rearrangement gives $[\mu]_{(A_i)_{i \in I}} \otimes [\rho]_{(B_k)_{k \in K}} = [\mu \oplus \rho]_{(A_i)_{i \in I} \times (B_k)_{k \in K}}$.*

- *Composition with another rearrangement gives $[\mu]_{(A_i)_{i \in I}} \, \mathring{\varsigma} \, [\nu]_{(A_{\mu(j)})_{j \in J}} = [\nu \, \mathring{\varsigma} \, \mu]_{(A_i)_{i \in I}}$.*

*If $\mu$ is natural, then the rearrangement applied on the product of a family of morphisms $(f_i)_{i \in I}$ where $f_i : A_i \to B_i$ yields;*

$$\left( \prod_{i \in I} f_i \right) \, \mathring{\varsigma} \, [\mu]_{(B_i)_{i \in I}} = [\mu]_{(A_i)_{i \in I}} \, \mathring{\varsigma} \, \left( \prod_{j \in J} f_{\mu(j)} \right)$$

### 3.1 Implementing Categories

To implement a monoidal product category, we will rely on lone objects $L$ and root morphisms $M$. This loses the potential structure of expressing nested product objects. Owing to the associativity of the monoidal product, however, nested products are isomorphic to flat products and we do not lose generality (Joyal & Street, 1991; Wilson et al., 2024). In the implemented lone object product category, objects are generated from the flat product of predefined lone objects. Morphisms are generated from category specific root morphisms and rearrangements placed into product and compositional constructs. Finally, we have blocks, which serve as construction rules for functions $B : \mathcal{C}(A, B) \to \mathcal{C}(A, B)$ and therefore allow for loops and aesthetic tags to be present in expressions.

We now realize Definition 5 in diagrams and code using the construction scheme of Section 2. Product objects, composition, products of morphisms, and blocks are construction rules. Root morphisms and rearrangements are root terms. Domains, codomains, and rearrangement data are exposed by metadata and methods. The high-level scaffold is shared by all product categories; specific categories arise by choosing the root morphisms and the class of allowed and natural remappings.

- **Objects** Ob$\mathcal{C}$. Objects are terms which anchor composition. Every object is represented as a tuple of *lone objects* $A = \Pi_{i \in I} A_i$, together with the unit object $\mathbb{1} = \Pi_{i \in \emptyset}$. In formal diagrams, objects are drawn as wires with arrows.

```
1  class ProductObject[L]:
2      content: Prod[L]
```

- **Morphisms** Mo$\mathcal{C}$. Morphisms are composable terms with a *domain* and *codomain* object. These interfaces determine when sequential composition is valid, and in diagrams they appear to the left and right of a morphism.

```
1  abstract class Morphism[L]:
2      def dom() -> ProdObject[L]: ...
3      def cod() -> ProdObject[L]: ...
```

Compound morphisms are built from composition and products, ultimately terminating in seed morphisms equipped with metadata. At minimum, that metadata supplies the domain, codomain, and a symbolic or pictorial description of how the morphism acts.

- **(Sequential) Composition.** Composition combines morphisms horizontally whenever the codomain of one matches the domain of the next. The resulting term is again a morphism, and associativity means that only the order of morphisms matters, not the parenthesization.

```
1  class Composed[L, M: Morphism[L]]
2      (Morphism[L]):
3      content: Prod[M]
4      def dom(): return content[0].dom()
5      def cod(): return content[-1].cod()
```

- **(Parallel) Product.** The product of objects concatenates their lone object contents, while the product of morphisms concatenates the corresponding domains and codomains. In diagrams, this is vertical stacking.

```
1  class ProductOfMorphisms
2      [L, M: Morphism[L]]
3      (Morphism[L]):
4      content: Prod[M]
5      def dom(): return ProdObject(concat(
6          m.dom() for m in content))
7      def cod(): return ProdObject(concat(
8          m.cod() for m in content))
```

Products place morphisms in parallel, so bifunctoriality allows horizontal and vertical composition to be interchanged when shapes match. Because diagrammatic terms are built from $\mathbin{\mathsf{g}}$ and $\otimes$, this law is axiomatically enforced in diagrams (see Section 2.2.1).

- **Rearrangements.** Given a domain $A = \Pi_{i \in I} A_i$ and a mapping $\mu : J \to I$, we generate a rearrangement morphism $[\mu]_{(A_i)_{i \in I}} : \Pi_{i \in I} A_i \to \Pi_{j \in J} A_{\mu(j)}$.

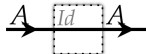

```
1  class Rearrangement[L](Morphism[L]):
2      mapping: Prod[int]
3      _dom: Prod[L]
4      def dom(): return ProdObject(_dom)
5      def cod():
6          return ProdObject(
7              _dom[mu_j]
8              for mu_j in self.mapping)
```

When the mapping is the identity, we generate an identity morphism.

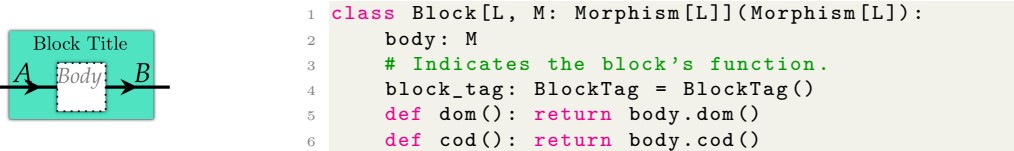

- **Blocks.** Blocks serve an organizational role in diagrams and code. They can distinguish function-ality, group subexpressions, or indicate repeated structure without changing the underlying type. They are diagrammed by placing a backdrop behind an expression.

```
1  class Block[L, M: Morphism[L]](Morphism[L]):
2      body: M
3      # Indicates the block's function.
4      block_tag: BlockTag = BlockTag()
5      def dom(): return body.dom()
6      def cod(): return body.cod()
```

Finally, this generates a recursive structure for defining the morphisms within a product category from the distinct root morphisms M. Hence, we have;

```
1  type ProductCategory[L, M: Morphism[L]] = (
2      M | Rearrangement[L]
3      | Composed[L, ProductCategory[L, M]]
4      | Block[L, ProductCategory[L, M]]
5      | ProductOfMorphisms[L, ProductCategory[L, M]]
6  )
```

## 4 Array-Broadcasted Category

### 4.1 Monoidal Arrays

The structure outlined in Section 3 will now be used to generate a template for arrays and broadcasted operations. This raises a number of challenges. We need a notion of lifting which generates reasonable objects and morphisms, respects the underlying monoidal rearrangement structure of the category, and allows for an algebra of lifted indexes. A naive approach would be to encode arrays $\mathbb{R}^n$ as the set of morphisms $\mathcal{C}(n, \mathbb{R})$ and to lift $f : A \to B$ via the hom-functor $\mathcal{C}(X, f) : \mathcal{C}(X, A) \to \mathcal{C}(X, B)$ which implements $g \mapsto g \mathbin{\mathring{\varsigma}} f$ for $g : X \to A$. This provides us access to useful algebra, including the Yoneda lemma, which allows us to work with rearranging data within arrays (see Section A.1). However, the hom-functor approach only applies to Cartesian closed categories i.e. copying being natural *and* hom-sets being self-contained. This raises issues as the set $\mathcal{C}(\mathbb{R}, \mathbb{R})$ is unmeasurable, making a probabilistic or computational interpretation intractable. Our notion of lifting, therefore, needs to use only the guaranteed monoidal rearrangement structure outlined above.

Instead of using the *compositional* nature of the underlying category to define arrays as in the hom-functor case, we choose to use the *monoidal product* nature of the underlying category to define arrays. Monoidal products support finite amalgamations of objects and morphisms, and guarantee that these always exist. For instance, the finite product of probability spaces $\Pi_{i \in I} X_i$ always generates a probability space, in a manner that does not apply to arbitrary hom-sets $\mathcal{C}(X, Y)$ as in this case of unmeasurable spaces such as $\mathcal{C}(\mathbb{R}, \mathbb{R})$. The limitations which propagate from this approach then reflect the limitations of the underlying algebra. Independent probabilistic operations run in parallel have characteristics distinct from independent

deterministic functions, for instance. This subsection presents a generic **monoidal array category** which extends a generic monoidal category with array and lifting structure, which will then be extended to the expressiveness needed for representing deep learning models.

### 4.1.1 Indexing Category

For a monoidal array category, we must first define the template for an indexing category used to manipulate the stride and shape of arrays. When it comes to representing deep learning models, this finite category will be restricted to finite affine transformations to enable stride to be reliably manage. By setting up the template, we ensure that the algebra holds at a high-level and may be extended in the future.

**Definition 7** (Indexing Category). *An indexing $\mathcal{I}$ category is a symmetric monoidal product category where we have;*

- ***Lone Objects*** $P_i$ *have a finite number of totally ordered **elements*** $\mathrm{El}(P) = \mathcal{I}(\mathbb{1}, P_i)$. *Objects* $P = \Pi_{i \in I} P_i$ *are built from the flat associative product of lone objects, and their total order ranks sequential elements. The elements of the product of objects is* $\mathrm{El}(P) = \{\Pi_{i \in I} p_i | \forall i \in I. p_i \in \mathrm{El}(P_i)\}$.

- ***Morphisms*** *between objects* $\eta : P \to Q$ *are uniquely identified by their action on elements, so that if* $p \, \mathring{,} \, \rho = p \, \mathring{,} \, \eta$ *for all* $p \in \mathrm{El}(P)$, *then* $\rho = \eta$ *i.e. morphisms are uniquely identified by their mapping from input to output elements.*

### 4.1.2 Monoidal Array Category

To extend a generic symmetric monoidal product category $\mathcal{C}$ to a parallelized monoidal array category $[\mathcal{C}; \mathcal{I}]$, we introduce a collection of synthetic array objects with shapes determined by an indexing category $\mathcal{I}$. These arrays objects are relabeling of the product of objects with themselves, so that $[X; P]$ for $X \in \mathrm{Ob}\,\mathcal{C}$ and $P \in \mathrm{Ob}\,\mathcal{I}$ corresponds to $\Pi_{p \in |\,\mathrm{El}\,|_P} X$. In this perspective, arrays are simply reannotations of already existing objects, no *new* mathematical structure is introduced. Rather, we develop a language and template to deal the common structure of parallelized operations, allowing this framework to be applied onto any symmetric monoidal category.

We will use a powerful tool from category theory called **canonical forms** (Wilson et al., 2024; Joyal & Street, 1991). In a category, objects $A$ and $A'$ are isomorphic if there exists a pair of inverse morphisms $\eta' : A \to A'$ and $\eta : A' \to A$. Critically, this enables us to choose $A$ as the canonical form of the objects. Morphisms $f : B \to A'$ to $A'$ are replaced with $f \, \mathring{,} \, \eta : B \to A$, while morphisms $g : A' \to C$ are replaced with $\eta' \, \mathring{,} \, g : A \to C$. This preserves composition, as $f \, \mathring{,} \, g : B \to C$ becomes $f \, \mathring{,} \, \eta' \, \mathring{,} \, \eta \, \mathring{,} \, g = f \, \mathring{,} \, g$. Usefully, this process removes all mention of $A'$ from the category without loss of generality. The distinction between isomorphic objects $A$ and $A'$ is, in essence, a mere symbolic tag to guide composition.

**Definition 8** (Isomorphism and Canonical Forms). *In a category $\mathcal{C}$, objects $A$ and $A'$ are **isomorphic** if there exists two morphisms $\eta' : A \to A'$ and $\eta : A' \to A$ so that $\eta \, \mathring{,} \, \eta' = \mathrm{Id}_{A'}$ and $\eta' \, \mathring{,} \, \eta = \mathrm{Id}_A$.*

*Given isomorphic objects $A$ and $A'$ and chosen canonical isomorphisms $\eta : A' \to A$ and $\eta : A' \to A$, we can compress their expression into a canonical form yielding the new category $\mathcal{C} \bmod \langle A', \eta, \eta' \rangle$ by using a functor (a composition preserving map) $\bmod \langle A', \eta, \eta' \rangle$ which maps objects $A' \mapsto A$, morphisms $f : B \to A'$ with $A'$ as the codomain to $f \, \mathring{,} \, \eta : B \to A$, and morphisms $g : A' \to C$ with $A'$ as the domain to $\eta' \, \mathring{,} \, g : A \to C$.*

We can then run the process in reverse to create **synthetic objects**. The involves taking a category $\mathcal{C}$ and introducing novel object symbols which are isomorphic to existing objects. The underlying meaning of morphisms in the new category $\mathcal{C}'$ can then be found by applying the synthetic canonical isomorphisms to yield an expression in $\mathcal{C}$. Nonetheless, $\mathcal{C}'$ differs in that composition is natively restricted – "decanonical" objects cannot be composed without reduction – and reveals the algebra of consistent structures which may not otherwise be apparent. We can now introduce monoidal arrays [X; P] as synthetic objects which reexpress the parallel product of an underlying object with itself, introducing arrays by using the guaranteed mathematics of the underlying category.

**Definition 9** (Monoidal Array Category). *Given an underlying base symmetric monoidal product category $\mathcal{C}$ with a lone object form (i.e. product objects are decomposable into singular root objects) and an indexing*

category $\mathcal{I}$ which conforms to the allowed rearrangements of $\mathcal{C}$, we construct $[\mathcal{C};\mathcal{I}]$ by introducing synthetic objects called **arrays** $[X; P]$ with $X$ taken from $\mathcal{C}$ and $P$ taken from $\mathcal{I}$.

Arrays $[X; P]$ are isomorphic to $\Pi_{p\in|\mathrm{El}(P)|}X$ by a pair of isomorphisms called the **join** $\mathrm{Jn}(X, P)$ : $\Pi_{p\in|\mathrm{El}(P)|}X \to [X; P]$ and **separator** $\mathrm{Sp}(X, P) : [X; P] \to \Pi_{p\in|\mathrm{El}(P)|}X$.

With the language provided by the monoidal array category, we can introduce batch lifted morphisms to express parallelization and reindexing morphisms to manipulate array stride. Batch parallelized morphisms are akin to hom-functors, but use the synthetic array objects rather than hom-sets which may not be supported by the underlying category. Reindexing morphisms indicate stride manipulations – such as diagonalizing an axis or performing a convolution – by borrowing the rearrangement structure of the category. As the indexing category $\mathcal{I}$ is defined to be totally ordered within the elements of each of its objects, it can instruct the construction of an ordered, monoidal product in the underlying category.

**Definition 10** (Batch Lifting and Reindexing Morphisms). *Given a monoidal array category $[\mathcal{C};\mathcal{I}]$, we construct;*

- **Batch Lift** *The batch lift of a morphism $f : X \to Y \in \mathrm{Mo}([\mathcal{C};\mathcal{I}])$ by $P \in \mathrm{Ob}(\mathcal{I})$ is given by $[f, P; :][X; P] \to [Y; P]$ so that;*

$$[f, P] = \mathrm{Sp}(X, P) \mathbin{\mathring{\,}} \prod_{p\in|\mathrm{El}(P)|} f \mathbin{\mathring{\,}} \mathrm{Jn}(Y, P)$$

- **Reindexing Morphism** *Given a morphism $\eta : P \to Q$ of $\mathcal{I}$ and an object $X \in [\mathcal{C};\mathcal{I}]$, we define the reindexing morphism $[X; \eta] : [X, Q] \to [X; P]$ to give;*

$$[X; \eta] = \mathrm{Sp}(X, Q) \mathbin{\mathring{\,}} [\eta]_{(X)_{q\in|\mathrm{El}(Q)|}} \mathbin{\mathring{\,}} \mathrm{Jn}(X, P)$$

Notably, both of these definitions are compositional, and in certain cases are dually so, passing through each other. This resembles the Yoneda lemma, but takes into account the underlying algebraic structure of the category. This is shown in Appendix A.1.

**Lemma 1** (Compositional Properties of the Array Monoidal Category). *In an array monoidal category, we have $[f \mathbin{\mathring{\,}} g; P] = [f; P] \mathbin{\mathring{\,}} [g; P]$ along with $[X; \rho \mathbin{\mathring{\,}} \eta] = [X; \eta] \mathbin{\mathring{\,}} [X; \rho]$. When either $\eta : P \to Q$ is natural within the underlying category or $f : X \to Y$ is deterministic, obeying naturality for all rearrangements, then we have $[f; Q] \mathbin{\mathring{\,}} [Y; \eta] = [X; \eta] \mathbin{\mathring{\,}} [f; P]$.*

For the purposes of diagrams and code implementation, we will use standard canonical forms of the lifted constructs. Much like flattening objects provides easier representation without loss of generality, these canonical forms of lifted constructs avoids mixing products in the base and lift of expressions. Note that the pseudocode here is to show correspondence to construction rules, and during implementation take a different form.

- **Array Objects.** Objects are drawn as an arrowed base object, with wires placed above to indicate lifting. This employs the canonical form $[[X; P]; Q] \mapsto [X; Q \otimes P]$ so that each new wire is placed at the top, and $X \mapsto [X; \mathbb{1}]$ (no wire) for consistency.

$$\frac{P}{\underset{A}{\underline{\qquad\qquad}}}\longrightarrow$$

```
1  class Array[B, A]:
2      base: B
3      shape: ProdObject[A]
4  def object_lift(base: Array[B, A], lift:
       ProdObject[A]):
5      return Array(base.base, lift + base.shape)
6
```

- **Products.** As unmarked vertical stacking corresponds to lifting, products must be distinguished by a dashed wire. Furthermore, to avoid mixing products, we enforce the rule that $[\Pi_{i\in I}X_i; P] \mapsto \Pi_{i\in I}[X_i; P]$. This requires the underlying category $\mathcal{C}$ to be symmetric.

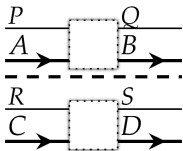

- **Batch Lift.** The batch lift of morphisms is indicated by placing a wire over an expression. This properly adjusts the domain and codomain shapes. In diagrams, batch lifting composition is axiomatically enforced.

```
1  class BatchLift[B, A, M: Morphism[Array[B,A]]](
      Morphism[Array[B,A]]):
2    base: M
3    lift: ProdObject[A]
4    def dom() -> object_lift(base.dom(), lift)
5    def cod() -> object_lift(base.cod(), lift)
6
```

- **Reindexing.** Reindexings are placed on the relevant wires, distinguished from base morphisms by being drawn as hexagons. Note how the order of the domain and codomain reverse. In diagrams, reindexing composition is axiomatically enforced.

```
1  class Reindexing[B, A, S: Morphism[A]](Morphism[
      Array[B,A]]):
2    base: B
3    stride: S
4    def dom(self) -> Array(base, stride.cod())
5    def cod(self) -> Array(base, stride.dom())
6
```

Diagrams allow for the combined compositionality of batch lifting and reindexing to be shown diagrammatically as a sliding operation as below if $f$ and $\eta$ are mutually natural;

Similarly to a the `ProductCategory` construct, this yields a recursive code implementations of a generic array monoidal category.

```
1  type ArrayProductCategory[B, A, M: Morphism[Array[B,A]], S: Morphism[A]] = (
2    ProductCategory[
3      Array[B,A],
4      M | Reindexing[B,A,S] | BatchLift[B,A,M]
5    ])
6
```

### 4.1.3  Weaves and Integrated Reindexings

The outline above provides the full mathematical basis for representing deep learning models. Already, we can generate deep learning models by setting the indexing category $\mathcal{I}$ to be affine stride transformations with axes (and their product) as objects and the underlying category $\mathcal{C}$ to represent datatypes and morphisms between them. These morphisms can be deterministic functions or probabilistic Markov kernels, the foundational template is the same. Arrays, parallelization, and stride transformations are then imposed using the template above. A graph representation follows from us working with a symmetric monoidal category (Piedeleu & Zanasi, 2025), and this graph representation can consider arrays as whole objects

rather than a mere concatenation of individual values. We can then layer the **Para** construct over the array monoidal representation to consider parameterized learned layers and backpropagation.

However, we want to introduce one more layer of features to better model how deep learning architectures are used in practice. With the tools provided so far, an operation targeting the top axis may be represented as below. On the right, we show the expression without the base object/underlying datatype, as it is often secondary to understanding an architecture.

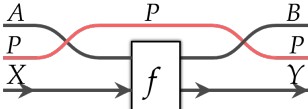
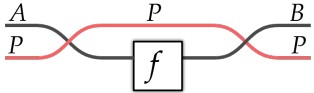

Though mathematically clear, this misses important details. In practice, the operation will be computed over the top axis without necessarily executing a stride transformation, which may incur computational cost. Computing matrix multiplication using tensor cores may be sensitive to the shape of inputs in a way that is not captured if indexing information is moved outside the core morphism. Additionally, these diagrams and representations lack conciseness. Instead, we wish to represent this operation as in the form below, integrating these local manipulations into the expression itself.

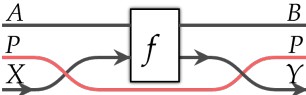
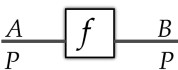

To integrate indexing information into the morphism itself, we use **weaves**. Weaves indicate which of the indexing objects form part of the **tile** or **target** of an expression. This encodes a symmetric remapping. We incorporate weaves for the codomain segments and weaves accompanied by reindexings for the domain segments. This avoids the problem of indexing information being encoded separately from the operation itself, allowing our representation to show architectures in a concise and computationally meaningful manner. This is shown in Figure 1, Listing 4.1.3, and Defintion 11.

Figure 1: An example of a broadcasted operation. Weaves use tags to rearrange data. Starting from the right, codomain weaves organize index objects into the degree, which is used to lift an underlying operation. For each input domain segment, a reindexing is applied over the degree. The result is then organized using the domain weaves. Typically, base objects are of secondary importance. However, structurally significant base objects may be highlighted as in the case of $n$.

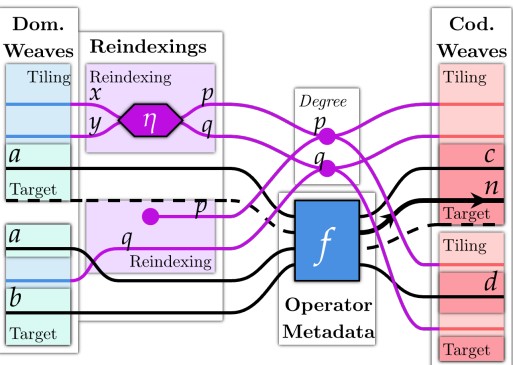

**Definition 11** (Weaves and Broadcasted Operations). *A **weave** $w$ over an indexing object $\Pi_{\ell \in L} P_\ell$ in a symmetric indexing category $\mathcal{I}$ tags each of $\ell \in L$ inputs to be sent to the front (tiled) or back (target). The front $\alpha_w : L \to F_w$ remapping extracts front objects, and the back remapping $\beta_w : L \to B_w$ extracts the back objects. Together, $\sigma_w = [0,0]_L \, \r{\circ}_9 \, (\alpha_w \otimes \beta_w)$ forms a symmetric map.*

*A broadcasted operation in $[\mathcal{C}; \mathcal{I}]$ has;*

- *A **target morphism** $f : \Pi_{i \in I}[X_i; A_i] \to \Pi_{j \in J}[Y_j; B_j]$,*

- *A **degree** $P \in \mathrm{Ob}(\mathcal{I})$,*

- *A $J$-family of **codomain weaves** $(c_j)_{j \in J}$, each of which has $F_{c_j} = \mathrm{len}\, P$ and $B_{c_j} = \mathrm{len}\, B_j$,*

```python
1  class Operator: ...
2  class Weave[B, A]:
3    base: B
4    shape: Prod[A | TILED]
5    def impose(tiling: ProductObject[A]):
6      return Array(base, ProductObject(
7        next(tiling) if a == TILED else a
8        for a in shape))
9  class Broadcasted[B, A, S: Morphism[A]](Morphism[Array[B,A]]):
10   target_operation: Operator
11   dom_weaves: Prod[Weave[B, A]]
12   cod_weaves: Prod[Weave[B, A]]
13   reindexings: Prod[ProductCategory[S]]
14   def dom(): return ProductObject(
15     weave_i.impose(reindexing_i.cod)
16     for weave_i, reindexing_i in zip(dom_weaves, reindexings)
17   )
18   def degree(): return allequals(
19     reindexing_i.dom for reindexing_i in reindexings_i
20   )
21   def cod(): return ProductObject(
22     weave_j.impose(degree())
23     for weave_j in cod_weaves
24   )
```

Listing 1: To avoid layered broadcasts, we use a generic operator tag to indicate the underlying operation. Weaves serve the role of indicating the target arrays.

- *An $I$-family of **reindexing** morphisms $(\eta_i)_{i \in I}$ from $\mathrm{Mo}(\mathcal{I})$ where $\eta_i : P \to Q_i$,*

- *And an $I$-family of **domain weaves** $(d_i)_{i \in I}$, each of which has $F_{d_i} = \mathrm{len}\, Q_i$ and $B_{d_i} = \mathrm{len}\, A_i$.*

*This then forms an operation $B$ mathematically equivalent (though may vary computationally from) (see Section 4.2);*

$$B : \prod_{i \in I}[X_i; \mathrm{cod}\,[\sigma_{dj}]_{(Q_i, A_i)}] \to \prod_{j \in J}[Y_j; \mathrm{cod}\,[\sigma_{cj}]_{(P, B_j)}]$$

$$B = \left(\prod_{i \in I}[X_i; (\eta_i \otimes \mathrm{Id}\, T_i) \mathbin{\S} [\sigma_{di}^{-1}]_{(Q_i, A_i)}]\right) \mathbin{\S} [f; P] \mathbin{\S} \left(\prod_{j \in J}[Y_j; [\sigma_{cj}]_{\mathrm{cod}\,[\sigma_{cj}]_{(P, B_j)}}]\right)$$

## 4.2 Imposing Structure – The $[\mathcal{B}; \mathcal{A}]$ Category

We now use the template above to define a mathematical description of deep learning models. This will be done by multiple rounds of layering templates on top of each other, yielding a rich description with each level of abstraction sharing key structural properties. These layers can be adjusted in the future to accommodate features outside of the scope of this work, such as considering differentiability.

We define $\mathcal{B}$ to be **BorelStoch** (Fritz et al., 2023), the category of standard Borel spaces and measurable Markov kernels. Borel spaces correspond to the datatypes we need, real numbers, complex values, or finite integers, equipped with a notion of probability. In **BorelStoch**, morphisms are Markov kernels, with each $f : X \to Y$ propagating elements of $X$ to probability distributions $P_f(y|x)$ over $Y$. Measurable deterministic functions are realized as generating Dirac distributions $P_f(y|x) = \delta_{f(x)}(y)$ The product is the product of $\sigma$-algebras, places morphisms in independent parallel, and supports naturally symmetric swapping.

**Definition 12** (Base Category – **BorelStoch**). *The base category $\mathcal{B}$ is **BorelStoch**, a symmetric monoidal category with;*

- ***Objects.** Each $X \in \mathrm{Ob}(\mathcal{B})$ correspond to standard Borel spaces $(X, \Sigma_X)$. Objects are decomposable into lone objects, whose $\sigma$-algebras are isomorphic to $\mathbb{R}$, $\mathbb{Z}$, or a finite discrete set $n \in \mathbb{N}$.*

- **Morphisms.** *Morphisms $f : X \to Y$ correspond to Markov kernels, mapping each $x$ in the set $X$ to a probability distribution $P_f(y|x)$ over $P(Y)$ in a measurable manner. Composition is given by chaining Markov kernels,*

$$P_{(f \, \mathbin{\mathchoice{\vcenter{\hbox{$\scriptstyle\circ$}}}{\vcenter{\hbox{$\scriptstyle\circ$}}}{}{}} g)}(z|x) = \int_Y P_g(z|y).dP_f(y|x)$$

- **Products.** *The product of objects $X \otimes Y$ corresponds to the Borel $\Sigma_{X \times Y}$ space over $X \times Y$ which contains all $\sigma_X \times \sigma_Y$ for $\sigma_X \in \Sigma_X$ and $\sigma_X \in \Sigma_Y$. The product of morphisms implements their kernels in independent parallel.*

In implementations, we do not define these exact maps. Rather, the operator tags of broadcasted constructs (see Section 4.1.3) indicate this operation. The aim of our implementation is to represent the structure of architectures. Evaluation can be viewed as a non-core property (see Section 2.2.1). Operations can be equipped with metadata which allows correct evaluation to be deduced and utilized by packages such as PyTorch or low-level non-structural mathematical analysis.

The base category is extended to arrays and parallelized operations with an indexing category $\mathcal{A}$ which has axes as objects and finite, affine transformations as morphisms. This class of operations supports standard stride transformations, axis rearrangements, and can be composed in a reliable manner. This restriction may be loosened, but this would make it difficult to track how axes interact with each other.

**Definition 13** (Indexing Category – Finite Affine Transformations)**.** *The indexing category is supplied by the category $\mathcal{A}$ of axes and finite affine transformations. It has;*

- **Objects.** *The lone objects $P_i$ of the category are **axes**, with elements $\mathrm{El}(P_i)$ corresponding to natural numbers $\{0, 1, \ldots, |P_i| - 1\}$. Objects are built from the product of these affine spaces, and correspond to shapes.*

- **Morphisms.** *Morphisms $\eta : P \to Q$ are finite affine transformations characterized by a linear map $\Lambda_\eta \in \mathbb{N}^{P \times Q}$ and an offset $v_\eta \in \mathbb{N}^Q$ so that elements $\boldsymbol{x} \in P$ are mapped according to;*

$$\eta(\boldsymbol{x}) = \Lambda_\eta \cdot \boldsymbol{x} + v_\eta$$

  *We impose the restriction that all these elements must be contained within $Q$. Composition is then given by;*

$$\Lambda_{\eta \, \mathbin{\mathchoice{\vcenter{\hbox{$\scriptstyle\circ$}}}{}{}{}} \rho} = \Lambda_\rho \cdot \Lambda_\eta \qquad v_{\eta \, \mathbin{\mathchoice{\vcenter{\hbox{$\scriptstyle\circ$}}}{}{}{}} \rho} = v_\rho + \Lambda_\rho \cdot v_\eta$$

We can now use the lifted array monoidal category $[\mathcal{B}; \mathcal{A}]$ to characterize the mathematics of deep learning architectures. This framework provides a language for describing probabilistic operations, parallelized maps, and demarcated arrays. This provides useful insight. For example, parallelizing a random operation such as dropout is defined to represent the operation performed in parallel with distinct random behaviour across each array index. Critically, this approach allows us to configure the size of axes and use polymorphic functions. As covered in Section 2.4, our implementation allows axes sizes to be pending terms. Therefore, we can generate expressions organized along demarcated arrays with pending axes sizes which can be assigned a static value or left open to support variable size inputs.

Next, we use **Para** (Fong et al., 2019) to work with paramaterized operations. **Para** acts as a wrapper around a symmetric monoidal category, generating **Para**($[\mathcal{B}; \mathcal{A}]$) which modifies the interpretation of composition and morphisms to "hide" inputs.

**Definition 14** (**Para**)**.** *Given a symmetric monoidal category $\mathcal{C}$, the symmetric monoidal category **Para**($\mathcal{C}$) has;*

- **Objects.** *The objects of **Para**($\mathcal{C}$) are the same as those of $\mathcal{C}$.*

- **Morphisms.** *A morphism $\langle \theta, f \rangle : X \to Y$ in **Para**($\mathcal{C}$) corresponds to an object $\theta$ and morphism $f : \theta \otimes X \to Y$ in $\mathcal{C}$ so that;*

$$\langle \theta, f \rangle \, \mathbin{\vcenter{\hbox{$\scriptstyle\circ$}}}_9 \, \langle \varphi, g \rangle = \langle \varphi \otimes \theta, (\mathrm{Id}\, \varphi \otimes f) \, \mathbin{\vcenter{\hbox{$\scriptstyle\circ$}}}_9 \, g \rangle$$

> *In this manner, **Para**($\mathcal{C}$) "hides" parameterized objects while maintaining composition. Hidden objects are accumulated into a tape, preventing them from disturbing desired compositional structure.*

By working with **Para**($[\mathcal{B}; \mathcal{A}]$) as our core mathematical definition, operations such as learned linear layers can be used without their additional weight inputs interfering with composition. It also leaves open the ability to extend our representation to natively consider differentiation and backpropagation by modifying $\mathcal{B}$ to support differentiation, though this is outside the scope of this work.

This core mathematical definition serves as a template for architectures. Within an expression, each constituent operation, rearrangement, block, product and composition corresponds to a precise mathematical map from inputs to outputs. Finally, we layer on an algorithmic interpretation, **Alg**. An algorithmic expression is open to further structure under the requirement that each component maps to a mathematical operation via a structure-preserving map (a functor) $F : \textbf{Alg} \rightarrow \textbf{Para}([\mathcal{B}; \mathcal{A}])$. The algorithmic expression, then, supports notions such as stride transforms being computationally costly, or datatypes being implemented at some level of quantization. This restricts the algebra of expressions – introducing unnecessary rearrangements such as $[X; [1, 0]_{(Q,P)}] \, \mathbin{\mathchar'40} \, [X; [1, 0]_{(P,Q)}]$ cannot be reduced to $\text{Id} [X; P \otimes Q]$, meaning in **Alg** we may have terms $f_0 \neq f_1$ even if $F(f_0) = F(f_1)$. This is critical, however, as optimizing implementation details lies in exploring mathematically-equivalent yet algorithmically-distinct expressions, which we may indicate as $f_0 \equiv f_1$. Furthermore, working in the **Alg** abstraction layer allows us to distinguish between pure product operations $f \otimes f$ and parallelized operations $[f; P]$. As arrays may be dynamically sized and provide guarantees related to parallelized computation, treating these two expressions distinctly allows for architectural and computational analysis.

The exact details of what constitutes **Alg** and how this can be developed to create a model of deep learning computation and optimization is outside the scope of this work. For example, it may involve implementing quantized datatypes by coupling an underlying datatype $\mathbb{R}$ with a tag $q$ indicating its quantization. The behaviour of this quantized datatype can then be analyzed, while mathematically interpreted as $\mathbb{R}$. This allows quantization information to be considered, while still preserving the notion that an operation executed at two distinct quantization levels is in some sense "equivalent". For now, it is sufficient to consider that the diagrammatic and code expressions of architectures generated by our framework correspond in "reality" to this additional layer over core mathematical definitions.

## 5 Key Operations

Now, we finally move onto diagramming and representing deep learning architectures, culminating in representing DeepSeek-V3 (DeepSeek-AI, 2025). All the mathematics and structure laid out above allows the definition of integrated broadcasted operations of Section 4.1.3 to serve as the basis for describing the components used by deep learning models by generating expressions in **Alg** tied to mathematical expressions in **Para**($[\mathcal{B}; \mathcal{A}]$). To represent various operations, we need to define operator tags. Typically, operators will be diagrammed by some pictogram, allowing for instance an expanding triangle for SoftMax. These typically correspond to polymorphic operations which may defined over axes of various sizes – such as SoftMax taking $\mathbb{R}^n \rightarrow \mathbb{R}^n$ defined over any $n \in \mathbb{N}$ – which can be configured according to the UID logic (see Section 2.4). These are then broadcasted using the weaving methods set up previously.

Indeed, when the underlying category supports natural deletion, as in the case of $[\mathcal{B}; \mathcal{A}]$ based on **BorelStoch**, then the broadcasting process corresponds to translating slices over the broadcasted axis. Slices correspond to Pythonic operations $[i, :]$, extracting specific rows or columns of an array. These map to reindexings generated by $[X; i \otimes \text{Id} \, Q] : [X; PQ] \rightarrow [X; Q]$ for $i \in \text{El}(P)$. This translation process is shown below in Figure 3.

Figure 2: SoftMax over the second-last dimension of an array can be shown with rearranging wires. The weaving of the operation indicates that it supplies $F(\mathbf{x})[p_0, :, p_1] = \text{SoftMax}(\mathbf{x}[p_0, :, p_1])$.

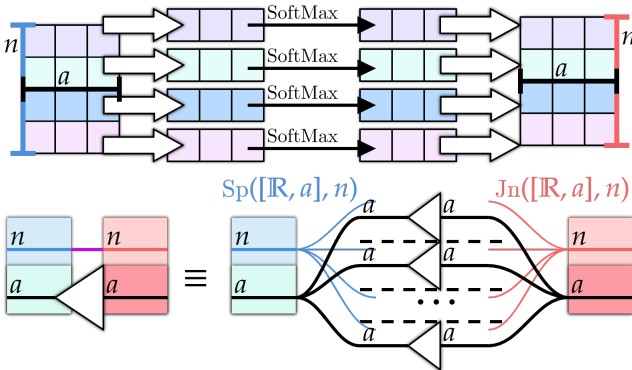

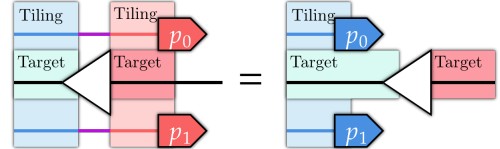

Figure 3: SoftMax over the second-last dimension of an array can be shown with rearranging wires. The weaving of the operation indicates that it supplies $F(\mathbf{x})[p_0, :, p_1] = \text{SoftMax}(\mathbf{x}[p_0, :, p_1])$.

## 5.1 Einstein Operations

Einstein operations are those which can be readily shown with the Einstein summation convention, and include transposes, summations, outer products, and inner products (Rogozhnikov, 2022). In the simplest case, we have an identity operation raised by a rearrangement, generating transposes, diagonalizations, and repetitions. The logic of these rearrangements can be seen through slice sliding, indicating clearly how inputs relate to inputs. These forms are diagrammed in Figures 4 and 5.

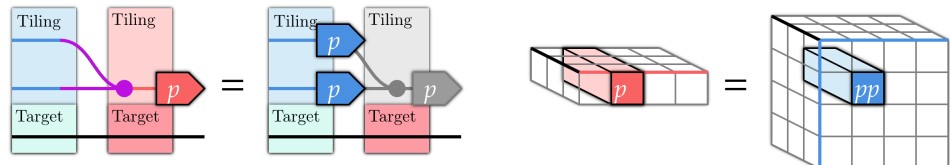

Figure 4: Diagonalization corresponds to the equation $\mathbf{y}[p, :] = \mathbf{x}[p, p, :]$. This can be expressed using the rearrangement reindexing $p \mapsto (p, p)$.

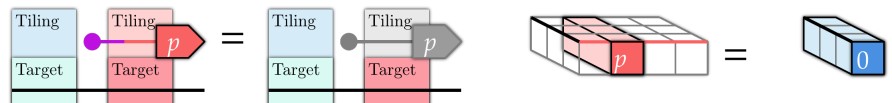

Figure 5: Repetition corresponds to the equation $\mathbf{y}[p, :] = \mathbf{x}[:]$. This can be expressed using the rearrangement reindexing $p \mapsto ()$.

Given the significance of matrix multiplication and summation operations, these also utilize the generic "pictogram-free" representation. A dashed line terminating implicitly yields a multiplication, $\mathbb{R} \otimes \mathbb{R} \to \mathbb{R}$. In Figure 6, the reindexings for the inputs are independent operations, and thereby we recover an outer product. Summation is indicated by a terminating wire or, if preceded by diagaonlization, a curved cup. The curved cup mimics Penrose graphical notation and provides visual distinctiveness to the critical dot product operation. This is shown in Figure 7. Thereby, the linear rearrangement logic of Penrose graphical notation or tensor networks (Biamonte, 2019) is recreated within our framework.

Figure 6: Multiplication can be expresed by having a dashed wire come to an end. The underlying operation is $(\cdot) : \mathbb{R} \times \mathbb{R} \rightarrow \mathbb{R}$, and therefore the target arrays $[\mathbb{R}, \mathbb{1}]$ have no axes. Here, with degree $P = P_0 P_1 P_2$, our reindexings are the rearrangements $p_0 p_1 p_2 \mapsto p_0 p_1$ and $p_0 p_1 p_2 \mapsto p_2$ respectively. This means the operation provides $\mathbf{z}[p_0, p_1, p_2] = \mathbf{x}[p_0, p_1] \cdot \mathbf{y}[p_2]$, or an **outer product**.

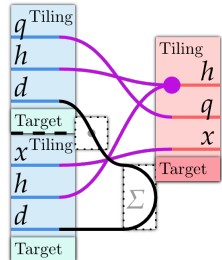

Figure 7: Weaved multiplication followed by summation yields an Einstein operation. Here, we show the operation 'q h d, x h d → h q x' which forms the query-key multiplication of multi-head attention. Note how the parallel calculation of heads is expressed by broadcasting.

## 5.2 Dropout – Randomness and Elementwise Operations

We can test the implications of **BorelStoch** as the underlying base category by assessing how dropout would be represented within our framework. Dropout is realized as a Markov kernel on elements, mapping $x \mapsto P_{\natural(r)}(x)$ which assigns a measure of $r$ to $\sigma_X \in \Sigma_X$ which contains $x$ and 0 to all Borel subsets which do not. Dropouts $\natural(r) : X \rightarrow X$ are fundamentally elementwise operations, which are broadcasted to forms such as $[\natural(r); P] : [X; P] \rightarrow [X; P]$. Using the slice sliding formulation, these correspond to $[\natural(r); P] \, \natural \, [X; i] = [X; i] \, \natural \, \natural(r)$, meaning each output result is evaluated independently. Therefore, all axes form part of the tiling rather than the target. If the datatype is implicit, corresponding to $\mathbb{R}$, the operation then "hovers" over an expression as shown in Figure 8.

Figure 8: Dropout expresses a probabilistic Markov kernel. The base operation maps $\natural : \mathbb{R} \rightarrow \mathbb{R}$ which is broadcasted over the appropriate axes. Thus, the resulting operation "hovers" in the diagram. This corresponds to an action on each element, as the $[\mathbb{R}; pq]$ output is determined by $\natural$ acting on the $[\mathbb{R}; pq]$ input. The mathematical meaning of this operation and its broadcasting behaviour is determined by $[\mathcal{B}; \mathcal{A}]$.

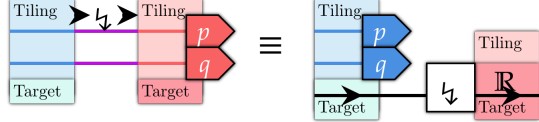

As our Yoneda sliding algebra only applies when either the underlying operation is deterministic (i.e. is natural with respect to all rearrangements) or reindexing is natural in the base category, repetitions do not slide over dropouts. Repetitions are generated by reindexings mapping $p \mapsto 0$, and therefore correspond to count increases with 0 being generated by every input. If we take the dropout over an axis then repeat that axis, it yields a distinct result to repeating that axis and applying droupout as diagrammed in Figure 9. This shows how the careful algebraic setup of Section 3 allows for a more accuarate understanding of the underlying algebra of expressions.

Figure 9: As dropout is non-deterministic – copying the result is not equivalent to copying the input and running the function in parallel – and repetition is non-natural, index sliding does not hold. This reflects the fact that repeating an axis full of generated random values is not the same as running the random operation in parallel across those axes.

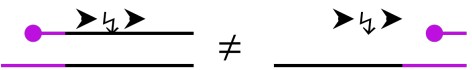

## 5.3 Learned Operations

Learned operations use stored weights to learn patterns within data. They can be categorically described using the **Para** construct. To indicate that an operation has a learned component, we bold some aspect of it. Diagramming linear layers and norming operations is shown in Figures 10 and 12. Note that our framework allows multidimensional learned layers to be clearly expressed, in contrast to PyTorch where multidimensional linear operations often require exogenous stride manipulation to properly manage.

Figure 10: A learned linear operation can be expressed by a chipped rectangle labeled **L**. The number and size of input/output axes is appropriately displayed according to the broadcasting framework. Here, we show the shape of the final heads aggregator layer of multi-head attention, with $\overline{x}$-indicating the number of tokens.

Figure 11: Under the **Para** framework, learned layers express a hidden input. We take an expression $[\mathbb{R}; \overline{x}hd] \otimes [\mathbb{R}; dhm] \to [\mathbb{R}; \overline{x}m]$, broadcasted over $\overline{x}$, and hide the second input. This allows composition to act as expected, and the series of input weights to be accumulated in a tape.

Figure 12: Norms are expressed as a bold (*learned*) circle with the inner space being occupied by a pictogram. The circle aims to be evocative of the idea of fitting values along a certain size. The "V" indicates RMSNorm.

## 5.4 Explicit Datatypes – Embeddings

Embeddings are noteworthy in that they are operations $\mathbf{E} : V \to [\mathbb{R}, m]$, where $V \in \mathbb{N}$ is the size of the vocabulary and $m \in \mathbb{N}$ is the token size. Therefore, the underlying datatype of their input array is an integer, and not a continuous value. This underlying datatype is then explicitly shown as an arrowed wire in diagrams. Given a $V$-sized datatype and a $V$-sized axis, we override the cupping notation to indicate the selection operation $V \times [\_; V] \to \_$. This allows us to show the internals of embedding as a selection operation in Figure 13.

Figure 13: The underlying operation of an embedding has shape $V \to [\mathbb{R}, m]$, where $V \in \mathbb{N}$. If we have made the real datatype implicit, then we need to explicitly label $V$ with an arrow. We can use selection to express the internals of embedding.

## 5.5 Mixture-of-Experts

This also provides the infrastructure to deal with Mixture-of-Experts. In a Mixture-of-Experts, we have linear layers for each of $n$ experts, realized as an additional $n$-input similar to embeddings. A specific token is inferenced by passing through a Top-K operation, which generates $k$ weights for the top values, and an array indicating the chosen expert indexes $[n; k]$. Each token evaluates its $k$ experts in parallel, at the end summing over the normalized top-k values. Using our framework, we can show the algebra of the GeGLU feed-forward experts used in DeepSeek-V3 (DeepSeek-AI, 2025) as in Figure 14. This can be understood as a step-by-step evaluation closely matching raw, Pythonic code.

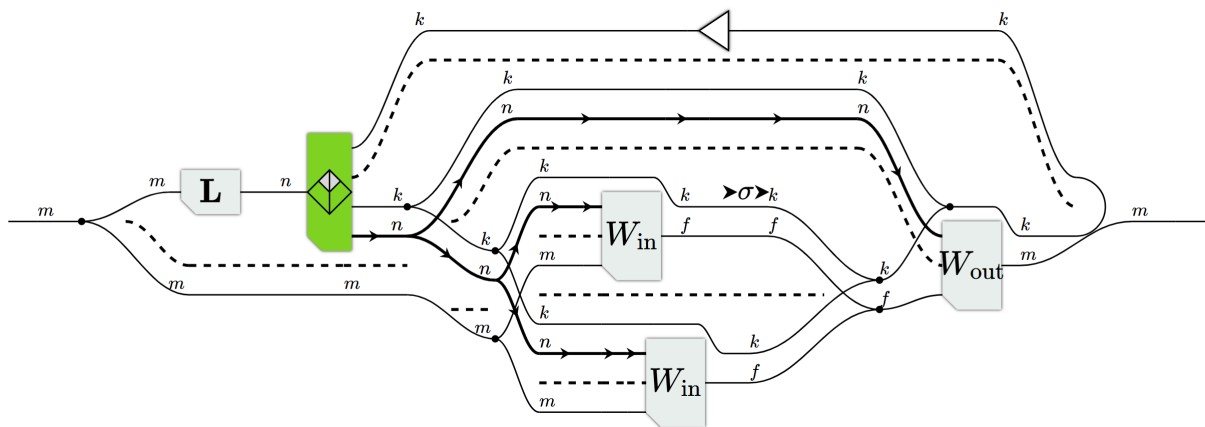

Figure 14: Here, we show a GeGLU mixture of experts as used in DeepSeek-V3 (DeepSeek-AI, 2025). We use a Top-K operation to select the indexes of the relevant arrays. The final output is then computed by summing over the normalized top-k weights, aligning with the chosen indexes.

Furthermore, we can use a mathematically simplified expression. In this form, we rearrange the $k$-index selection to occur in the outputs of linear layers. This leads to a cleaner, mathematically equivalent expressions, sans the "sparseness" of the axis. These "sparse" axes are introduced as a new axes types which references the local size at evaluation, $k$, relative to the total size, $n$, from which they are drawn. This is shown in Figure 15. Though this form is sufficient to mathematically express an architecture, it requires algebraic manipulation into Figure 14 to be evaluated. This algebraic manipulation can be supplied by the tools of Section 6.6.

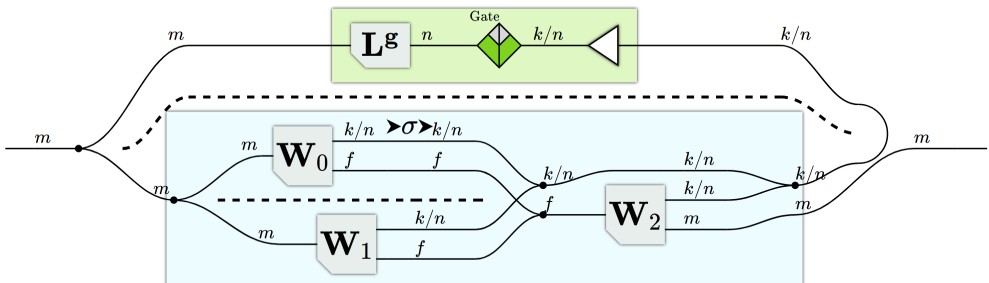

Figure 15: The expression from Figure 14 can be reexpressed in a mathematically equivalent form which rearranges indexed-inputs to outputs, over which we perform a sparse sum.

## 5.6 Convolution

Under our framework, convolution can be defined by a reindexing followed by a learned linear layer. Convolution can be pythonically expressed using slices as $\mathbf{y}[i_{\overline{x}'}, :] = \Sigma_{j_k \in k} \mathbf{L}[j_k] \cdot \mathbf{x}[i_{\overline{x}} + j_k, :]$. This can be split into a convolution operation which first reindexes $(*\mathbf{x})[i_{\overline{x}'}, j_k, \ell_c] = \mathbf{x}[i_{\overline{x}} + j_k, \ell_c]$ followed by a $kc$ to $c'$ linear layer $\mathbf{L}$. The convolution operator can be represented by a reindexing using addition, $+ : \overline{x}'k \to \overline{x}$. Because the convolution matrix can be supplied by a reindexing, it is computationally distinct from the general class of all matrices, and this distinction is clearly shown by our framework. Convolution expressed in our framework is shown in Figure 16.

A noteworthy benefit of our framework is the ability to reason about models. In the case of convolution, we are able to observe the translational equivariance by propagating indexes through the $\overline{x}'$ axis. We can define translation as a reindexing which shifts an element $i$ to $i + t$. We can "slide" this translation through the expression using the broadcasting rules and the fact that $(+)(i + t, k) = (+)(i, k) + t$. This sliding is derived

Figure 16: Convolution can be expressed as a two-step process of an addition reindexing (*convolution matrix*) followed by a learned linear layer.

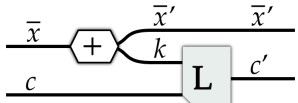

from the Yoneda sliding, elaborated in Appendix A.1. This is shown in Figure 17, revealing the translational equivariance of convolution across the $\overline{x}/\overline{x}'$ axis.

Figure 17: The equivariance of convolution can be shown diagrammatically by sliding an index-wise translation over the operation.

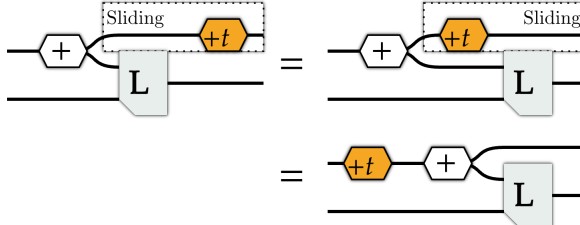

# 6 Results

Taken together, the tools presented above allows us to represent the entirety of the DeepSeek-V3 architecture (DeepSeek-AI, 2025). We can define it using the code outlined so far. This code expression grants us access to a "web" of automated features that allows us to manage this expression systematically. First, we will provide an outline of how a complex model such as DeepSeek-V3 is represented part-by-part. Then, we will cover how systematic code features – interoperable Python/TypeScript implementations (Section 6.2), autoalignment (Section 6.3), configuration generation (Section 6.4), hypergraph analysis (Section 6.6), and PyTorch compilation (Section 6.5) – work together to provide the utility needed to manage deep learning architectures as algebraic structures. This lays the foundation for future features, and an integrated web of tools for analyzing and managing deep learning models from a mathematical perspective (Section 7).

## 6.1 DeepSeek-V3 Representation

We begin with a representation of DeepSeek-V3, using the tools of Section 5. The code used to generate these representations is shown in Appendix B. To keep the representation manageable, we separate it into a series of blocks which indicate key regions. These blocks provides operators which can be decomposed at increasing levels of detail. At the highest level, we have a standard transformer-like layout: attention layers interspersed with feed forward layers, repeated some number of times. In this case, 27. Attention is provided by multi-head latent attention, where Einstein operations are used to define multi-head $Q - K$ matrix multiplication across a complex rotary and latent real component. The ability to represent various datatypes here is used to show complex variables with their distinct form of multiplication. The feed forward layers are supplied by GeGLU blocks (Shazeer, 2020), which use multiple linear layers in sequence. Finally, the Mixture-of-Experts layer uses the format outlined in Section 5.5.

## 6.2 Interoperable Code Package

The *DeepSeek-V3* model presented above is generated in Python, following an implementation which matches the term construction framework and the pseudocode presented alongside mathematical definitions. This results in functional expressions corresponding to the real terms of the algebra with UID terms being open to configuration and algebraic manipulation. These implementations are mirrored in TypeScript, where they are used to generate diagrams. Expressions can be encoded in JSON and sent between the packages, meaning we benefit from Python's familiarity and algebraic tools along with TypeScript's powerful rendering engine.

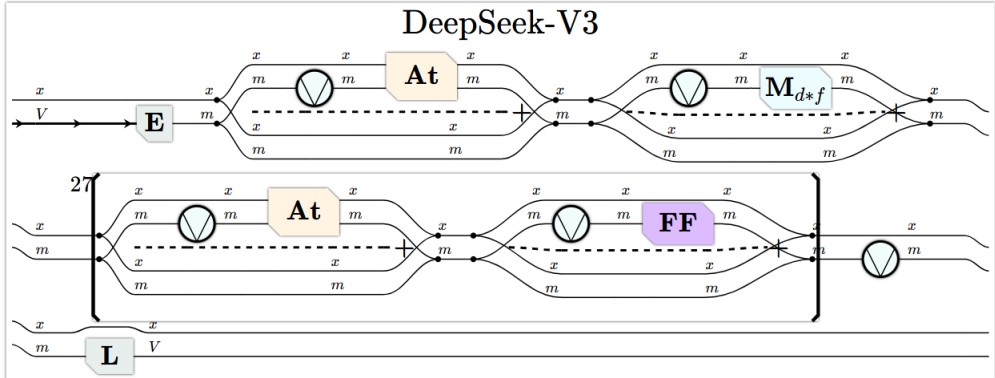

Figure 18: At a high-level, the deep seek algorithm follows a standard transformer structure: an embedding maps from a series of tokens indexed across $V \in \mathbb{N}$ options, and is sent through a sequence of attention-feed forward layers. The final result is mapped back through a linear layer, producing a vector $[\mathbb{R}; V]$ indicating the log-weights of output token options. These layers are all residual connections (He et al., 2015).

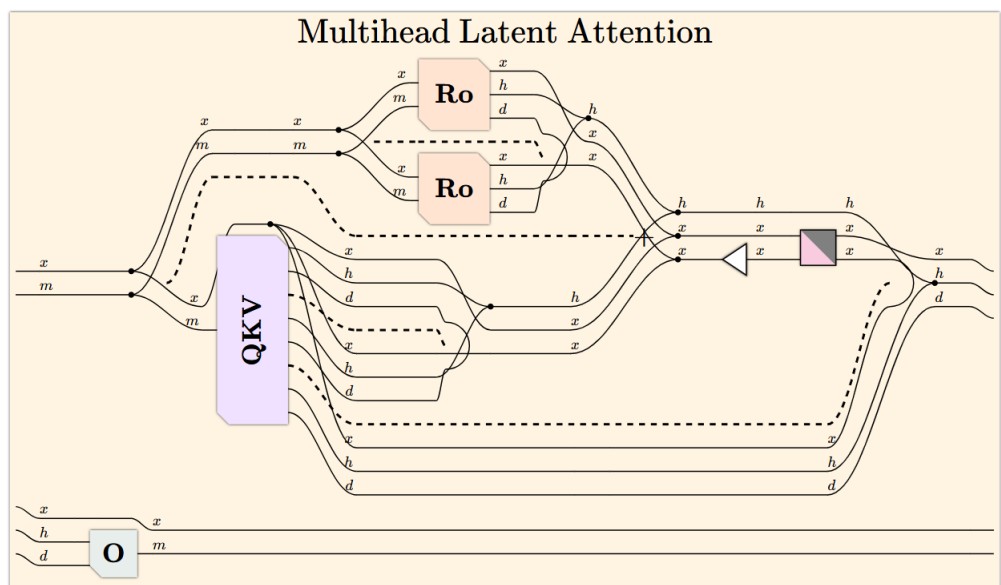

Figure 19: Attention blocks correspond to multi-head latent attention which comprises a complex rotary component and a real, latent component. These are used to generate queries and keys, which map through a multi-head $Q - K$ matrix multiplication contracted over the $d$ axes parallelized over $h$ heads. The results of the complex and real components are summed together, passed through a SoftMax and weighted lower triangular normalizer for causality, undergo an $S - V$ matrix muliplication, and are passed through an output linear layer, contracting the heads.

## 6.3 Autoalignment

A key difficulty with managing deep learning models is ensuring the size of axes properly match. This is important both for broadcasting to be clearly defined and for necessarily static axis sizes – such as those of linear layers – to be properly sized during implementation. In the Python implementation, we can use loose algebraic terms with pending UID axes which are then aligned during forced composition, indicated by @. This alignment can be seen as a *functor*, a composition preserving map between categorical expressions. Given $f : X_f \to Y_f$ and $g : Y_g \to Z_g$, we try to find functors $F_f$ and $F_g$ which maps the intermediate objects to a unified form, $F_f : Y_f \mapsto Y$ and $F_g : Y_g \mapsto Y$. Then, we work with $f@g = F_f(f) \, \tfrac{\circ}{\circ} \, F_g(g)$. This allows

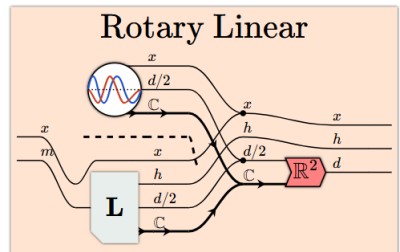

Figure 20: The rotary linear layer generates queries and keys for the complex component. This is supplied by a rotary mask generator, which is *not* "parallelized" over the $\overline{x}$ axis – the outputs are strictly dependent on which $\overline{x}$ index we are at. This "binds" the $\overline{x}$ axis into the output, meaning the result has positions encoded. This multiplies with a linear layer which generates both a real and complex value for the $m$ inputs. The resulting output is "decomplexed", and yields an array with integrated positional information.

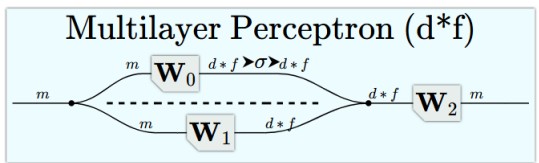

Figure 21: The multilayer perceptrons used by DeepSeek-V3 is a GeGLU (Shazeer, 2020). We have two input linear layers, one of which passes through an elementwise operation, the results multiplied together, and followed by an output linear layer. The internal size, in this case $d * f$, is not dependent on external axes sizes, and therefore can be configured independently.

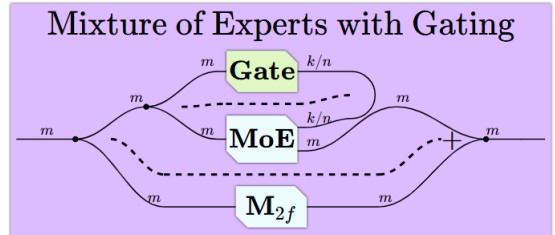

Figure 22: The Mixture-of-Experts block follows Section 5.5, with a gate selecting experts added to a multilayer perceptron. This architecture supports multi-GPU computational splits. Here, we show the simplified "sparse axis" presentation which presents a mathematically coherent representation insufficient for direct computation.

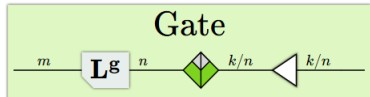

Figure 23: The gate for the Mixture-of-Experts involves a gate linear layer which provides weights for each of $n$ layers, a Top-K selection which generates a sparse axis, which is then normalized by a SoftMax-like operation, which, in practice, uses a sigmoid rather than an exponent to weight values.

expressions to be succinctly defined with properly aligned and recorded shapes. Autoalignment is not always possible, and tries to do the following process: if the number of axes in the top segment mismatch, batch lift an expression, then if the number of segments mismatch, take the product with an identity. This leads to two expressions with the same number of axes. Corresponding axes are then put in UID buckets with a canonical form and are applied over both expressions, ensuring alignment. This is shown in the table below and Figure 25, where templates for base expressions are composed to define attention.

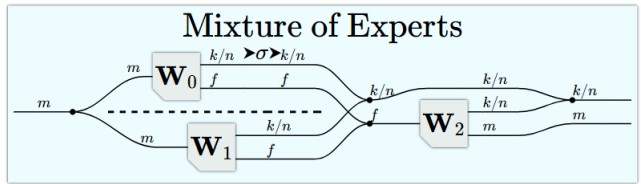

Figure 24: Finally, the experts component follows Section 5.5, with each layer having an additional $n$ axis of which $k$ values are non-zero.

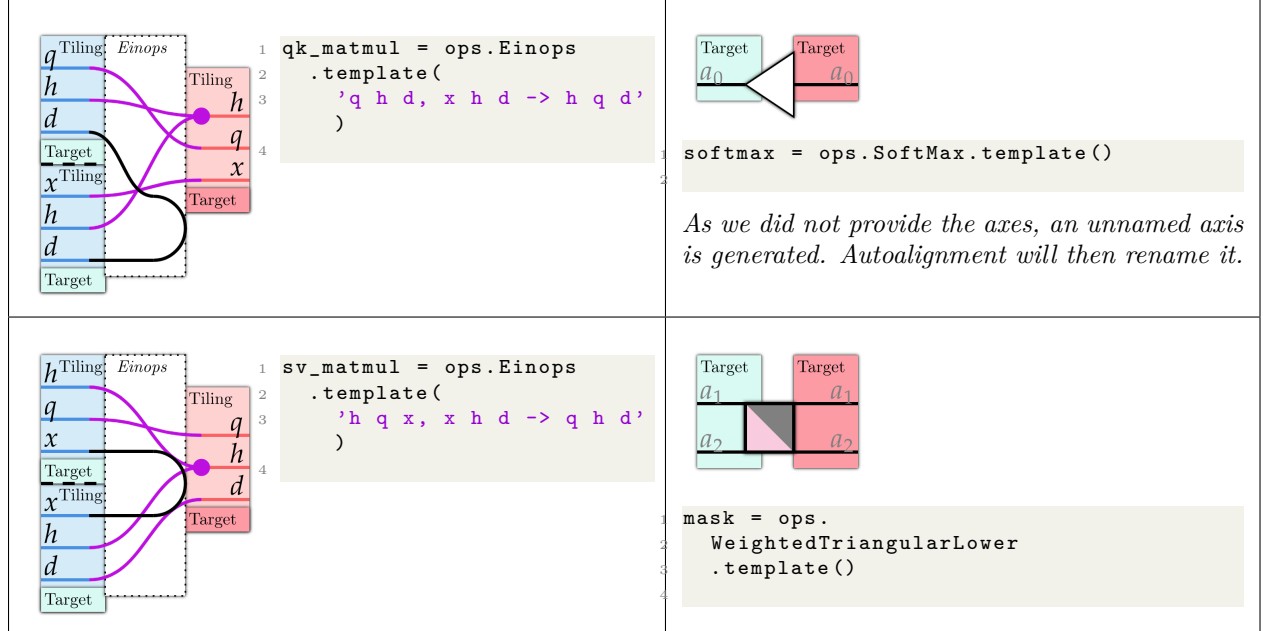

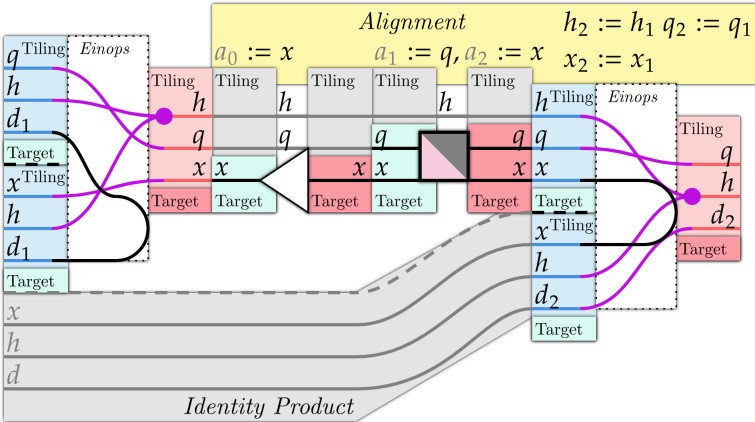

Figure 25: Applying `qk_matmul @ softmax @ mask @ sv_matmul`, we perform autoalignment operations at each step. Axes are aligned to be the same. Operations are batch broadcasted when the number of axes mismatch. In the case of `sv_matmul`, we take the product with an identity rearrangement with the array $[\mathbb{R}, xhd]$. Note that the $h$, $q$, $x$ axes of `qk_matmul` and `sv_matmul` are separately generated, meaning their equivalence is only realized after composition. The $d_1$ and $d_2$ axes are never aligned together, meaning that the final configuration of the expression takes $\langle q, h, x, d_1, d_2 \rangle$.

## 6.4 Configuration Generation

Constructed terms – such as the one expressed in Figure 25 – have free terms associated with UIDs. Above, we showed how axes can be automatically aligned during composition. The remaining UIDs reveals the overall degrees of freedom for configuration, and an expression can be scanned to find these terms. From the list of UIDs, we generate an assignment to set axes to desired sizes. In Figure 26, we provide the example of setting the axis sizes of multi-head attention placed inside a ResNet He et al. (2015), making the expression ready for constructing a PyTorch model.

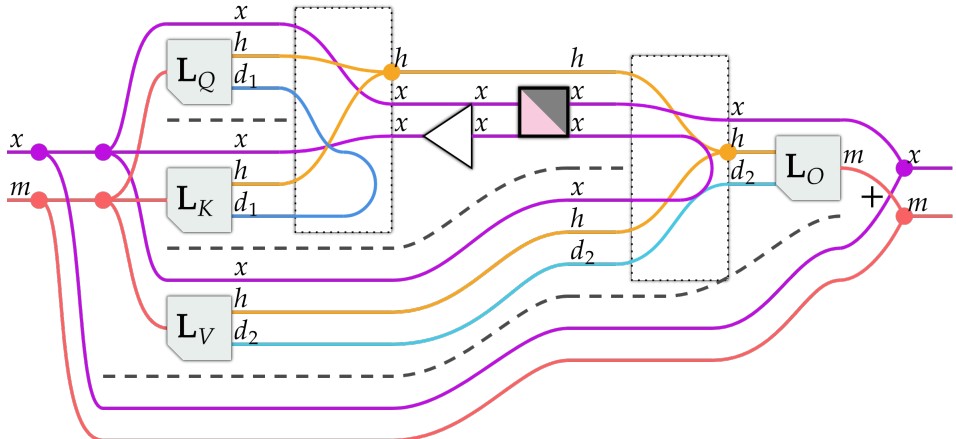

Figure 26: A full ResNet attention block expression constructed through autoalignment has a number of numeric sizes with UIDs. By assigning values to these, we have a configured expression ready for PyTorch compilation.

## 6.5 Compositional compilation to PyTorch

Algebraically defined models indicate execution instructions from start to end and, therefore, we have sufficient information to supply a corresponding PyTorch module. This can be done in a compositional manner by converting `Composed` constructs to sequential PyTorch modules, and `ProductOfMorphisms` to parallel PyTorch modules. Broadcasted operations supply execution metadata through the operator tag and broadcasting information in their weaves and reindexings. Due to PyTorch's elaborate broadcasting semantics, this conversion requires some infrastructure. The end result of this process allows us to take a configured constructed term, as above, and generate a runnable PyTorch module. Compilation to PyTorch is an incidental feature of the constructed terms. No aspect of the algebraic framework is explicitly defined for PyTorch. Compilation to any other framework – TensorFlow, Triton, or others – would all be done in a similar manner. Algebraic terms serve as a universal framework for compilation. An example of such a compilation is shown in Appendix B, where we compile a description of a multi-head transformer.

## 6.6 Algebraic Manipulation with Hypergraphs

A key feature of algebraic terms is algebraic manipulation, the ability to define general algebraic rules and apply these to models. The composed-product approach we outline above is tailored to constructing and diagramming terms. However, bifunctoriality and symmetry (see Section 3) yield redundancy. Certain algebraic properties are better captured by hypergraphs (Piedeleu & Zanasi, 2025) which discard immaterial information about morphisms' location among composed, product, and rearrangement structures. We can define an algorithm to convert from a `ProdCategory[L,M]` to a `Hypergraph[L,M]`, defining the conversion from mathematical first-principles and therefore having it applicable to any symmetric monoidal category, including **Para**$[\mathcal{B}; \mathcal{A}]$. A hypergraph form of Figure 26 is shown in Figure 27.

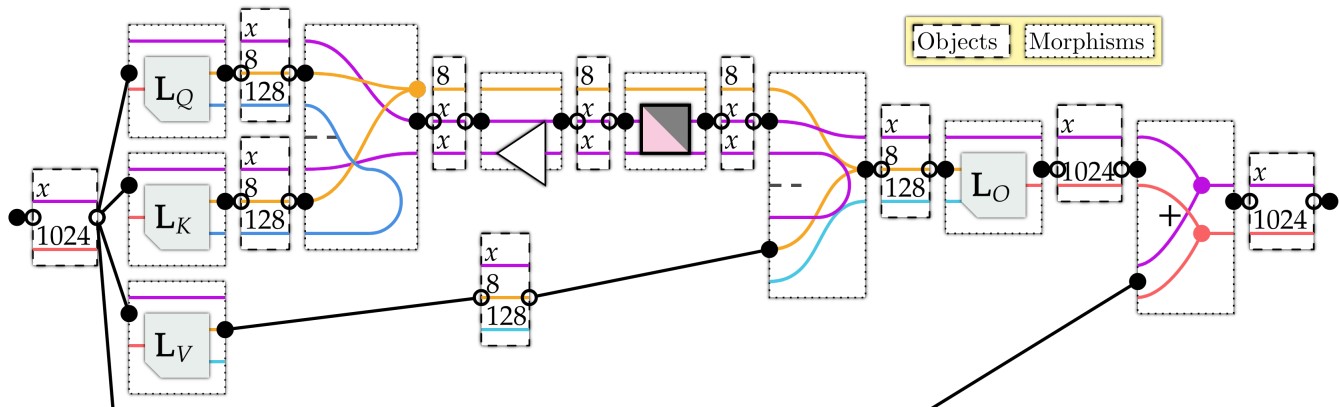

Figure 27: By converting from a "ProdCategory[L,M]" to a "Hypergraph[L,M]", we can perform algebraic manipulation with hypergraph rewrite rules. This allows us to apply general algebraic properties such as associativity, bifunctoriality, and symmetry, which are not easily captured by the composed-product approach.

### 6.7 Diagram Generation

Within TypeScript, we can implement the diagramming procedure. We convert `Composed` constructs into horizontally placed blocks, and `ProductOfMorphisms` constructs into vertically placed blocks. Objects are associated with a series of anchors which are sequentially linked together. The layered approach to defining our category allows us to begin with a framework for rendering categories in general, which is then specified for representing the category of deep learning architectures. In this manner, category theory's templating approach to mathematical structure allows code to be reused in different contexts. This diagramming tool has been used to generate diagrams of Mixture-of-Experts models (see Section 5.5) and *DeepSeek-V3* (see Section 6.1) and is shown in Appendix B.

## 7 Future Work and Conclusion

The formal mathematical framework established above and implementation following the construction rules paradigm allows us to process deep learning models algebraically. The dependency relations of implementations only have to follow the dependency relations of underlying mathematical definitions, allowing for modular code. This allows new features to be readily added.

The most impactful novel feature would be automatic low-level kernel derivation. This follows from the techniques outlined in **FlashAttention on a Napkin (FAN)** (Abbott & Zardini, 2025). That work would go beyond the scope of this paper, which focuses on the foundations of a formal approach. FAN is uniquely positioned to derive critical low-level optimizations which are inaccessible to standard compilations found in PyTorch. Tiled matrix multiplication and attention can be derived from first-principles. Furthermore, FAN allows for hardware-aware performance models, which guide improved future design. By integrating model analysis into mathematics, PyTorch, and pollable tools, AI-accelerated development of future algorithms will be within reach.

More broadly, various analyses dependent on compositionality will be possible within the categorical framework. The **Para** approach allows for backpropagated algorithms to be derived algebraically and piecewise. This offers an alternative to PyTorch's backpropagation tools, circumventing the need for its infrastructure almost entirely allowing for a true universal model of deep learning architectures which can be directly optimized and compiled into low-level code. Additionally, we can develop novel compositional analyses. A subject of particular interest is quantization error composition. Computational costs are linearly dependent on quantization size, while bandwidth costs – often the more pressing concern for models – are superlinearly dependent. Determining where to utilize lower quantizations is a byproduct of how effectively random

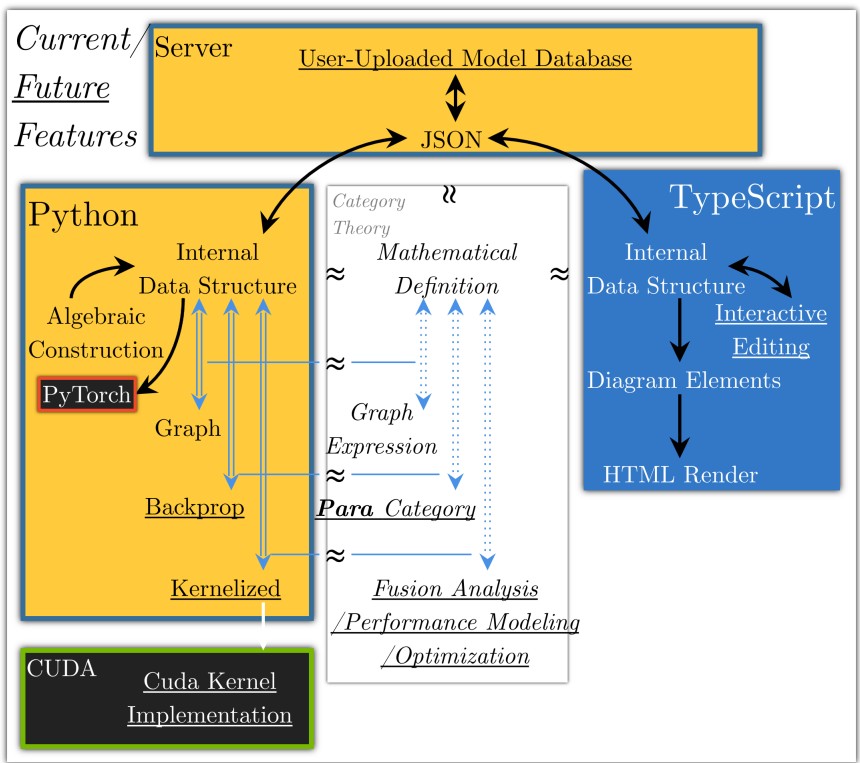

Figure 28: The modularity of the framework allows for a web of features to be developed and integrated. At any point, new features can be "hooked" into the system, allowing for a powerful and extensible framework for future AI research.

rounding errors will propagate. This is a compositional property, and therefore category theory's tools are of particular interest.

The diagramming tool can be developed into a suite of tools which allow for models to be developed and reasoned about diagrammatically, then shared to other uses through the natural JSON encoding resulting from the term construction system. Such a toolset will allow models to be intuitively manipulated and shared, enhancing research productivity.

This collection of modularity, mathematics, and tools leads to a "web" of features which can be extended into the future. At any point we can "hook" into the system (see Figure 28), creating a powerful framework for future AI research.

Finally, we can integrate the categorical and optimization tools of this framework into categorical codesign Zardini (2023) to optimize various stages of the artificial intelligence stack together. Models generate performance models, indicating the available memory-bandwidth-compute requirements for specified levels of performance, and this requirements-functionality relationship can be fit into the requirements-functionality relationships of hardware, power supply, and other components. This allows for a holistic approach to AI development, where the design of models, algorithms, and hardware are co-optimized.

Overall, the mathematical procedures, formal descriptions, algebraic implementation, and automated diagramming provided in this work lay the foundation for a robust, formalized, and systematic approach to the design of deep learning algorithms. This addresses a key challenge in the deep learning community, and opens the possibility of using artificial intelligence to design itself.

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

# A  Appendix

## A.1  Lifting Compositionality

*Proof.* For batch lifting, we use the join / split isomorphism pair along with bifunctoriality to compose products;

$$[f;P] \mathbin{\fatsemi} [g;P] = \mathrm{Sp}(X,P) \mathbin{\fatsemi} \prod_{p \in |\mathrm{El}(P)|} f \mathbin{\fatsemi} \mathrm{Jn}(Y,P) \mathbin{\fatsemi} \mathrm{Sp}(X,P) \mathbin{\fatsemi} \prod_{p \in |\mathrm{El}(P)|} g \mathbin{\fatsemi} \mathrm{Jn}(Y,P)$$

$$= \mathrm{Sp}(X,P) \mathbin{\fatsemi} \prod_{p \in |\mathrm{El}(P)|} f \mathbin{\fatsemi} \prod_{p \in |\mathrm{El}(P)|} g \mathbin{\fatsemi} \mathrm{Jn}(Y,P)$$

$$= \mathrm{Sp}(X,P) \mathbin{\fatsemi} \prod_{p \in |\mathrm{El}(P)|} (f \mathbin{\fatsemi} g) \mathbin{\fatsemi} \mathrm{Jn}(Y,P)$$

$$= [f \mathbin{\fatsemi} g; P]$$

For reindexing, we use the compositional property of rearrangements.

$$[X;\eta] \mathbin{\fatsemi} [X;\rho] = \mathrm{Sp}(X,Q) \mathbin{\fatsemi} [\eta]_{(X)_{q \in |\mathrm{El}(Q)|}} \mathbin{\fatsemi} \mathrm{Jn}(X,P) \mathbin{\fatsemi} \mathrm{Sp}(X,P) \mathbin{\fatsemi} [\rho]_{(X)_{q \in |\mathrm{El}(P)|}} \mathbin{\fatsemi} \mathrm{Jn}(X,R)$$

$$= \mathrm{Sp}(X,Q) \mathbin{\fatsemi} [\eta]_{(X)_{q \in |\mathrm{El}(Q)|}} \mathbin{\fatsemi} [\rho]_{(X)_{q \in |\mathrm{El}(P)|}} \mathbin{\fatsemi} \mathrm{Jn}(X,R)$$

$$= \mathrm{Sp}(X,Q) \mathbin{\fatsemi} [\rho \mathbin{\fatsemi} \eta]_{(X)_{q \in |\mathrm{El}(Q)|}} \mathbin{\fatsemi} \mathrm{Jn}(X,R)$$

$$= [X; \rho \mathbin{\fatsemi} \eta]$$

Finally, for the combined case we require that the condition below holds;

$$\left( \prod_{q \in |\mathrm{El}(Q)|} f \right) \mathbin{\fatsemi} [\eta]_{(Y)_{q \in |\mathrm{El}(Q)|}} = [\eta]_{(X)_{q \in |\mathrm{El}(Q)|}} \mathbin{\fatsemi} \left( \prod_{p \in |\mathrm{El}(P)|} f \right)$$

This can be ensured by either having $\eta$ be natural within $\mathcal{C}$, or $f$ being *deterministic*, treating all rearrangements as natural.

$$[f;Q] \mathbin{\fatsemi} [Y;\eta] = \mathrm{Sp}(X,Q) \mathbin{\fatsemi} \left( \prod_{q \in |\mathrm{El}(Q)|} f \right) \mathbin{\fatsemi} \mathrm{Jn}(Y,Q) \mathbin{\fatsemi} \mathrm{Sp}(Y,Q) \mathbin{\fatsemi} [\eta]_{(Y)_{q \in |\mathrm{El}(Q)|}} \mathbin{\fatsemi} \mathrm{Jn}(Y,P)$$

$$= \mathrm{Sp}(X,Q) \mathbin{\fatsemi} \left( \prod_{q \in |\mathrm{El}(Q)|} f \right) \mathbin{\fatsemi} [\eta]_{(Y)_{q \in |\mathrm{El}(Q)|}} \mathbin{\fatsemi} \mathrm{Jn}(Y,P)$$

$$= \mathrm{Sp}(X,Q) \mathbin{\fatsemi} [\eta]_{(X)_{q \in |\mathrm{El}(Q)|}} \mathbin{\fatsemi} \left( \prod_{p \in |\mathrm{El}(P)|} f \right) \mathbin{\fatsemi} \mathrm{Jn}(Y,P)$$

$$= \mathrm{Sp}(X,Q) \mathbin{\fatsemi} [\eta]_{(X)_{q \in |\mathrm{El}(Q)|}} \mathbin{\fatsemi} \mathrm{Jn}(X,P) \mathbin{\fatsemi} \mathrm{Sp}(X,P) \mathbin{\fatsemi} \left( \prod_{p \in |\mathrm{El}(P)|} f \right) \mathbin{\fatsemi} \mathrm{Jn}(Y,P)$$

$$= [X;\eta] \mathbin{\fatsemi} [f;Q]$$

$\square$

# B  Results Notebook

```
import asyncio
# Use server OR display locally, inline
USE_SERVER = True
import websocket_transfer.websockets_transfer as wst
import display as dsp
async def print_term(term):
    if USE_SERVER:
        await wst.send_term(term)
    else:
        dsp.print_category(term)
```

## Autoalignment

```
import construction_helpers as ch # Required for
overloaded operators
import data_structure.Category as cat
import data_structure.Operators as ops

qk_matmul = ops.Einops.template('q h d, x h d -> h q
x')
softmax = ops.SoftMax.template()
mask = ops.WeightedTriangularLower.template()
sv_matmul = ops.Einops.template('h q x, x h d -> q h
d')
attention_core = qk_matmul @ softmax @ mask @
sv_matmul
await print_term(attention_core)
```

```
Sending term to server...
Received from server: {"msgType": "Connected"}
Received from server: {"msgType": "DataReceived"}
```

*Diagram from tsncd,*

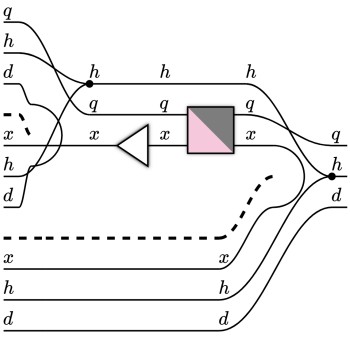

## Configuration Generation

```
import term_utilities.generate_config as gc
import display.display_config as dpc

## Making the multi-head attention block
## (0, 0) becomes a Rearrangement with mapping \mu:
0 ↦ 0, 1 ↦ 0
def resnet[B: cat.Datatype, A: cat.Axis](target:
cat.BroadcastedCategory[B, A]):
    return (0, 0) @ target @
ops.AdditionOp.template()

Lq, Lk, Lv = [ops.Linear.template('m', 2, name) for
name in ['q', 'k', 'v']]
```

```
Lo = ops.Linear.template(2, 'm', 'o')
attention_linears = (0, 0, 0) @ (Lq * Lk * Lv) @
attention_core @ Lo
attention_block = resnet(attention_linears)
await print_term(attention_block)
```

```
Sending term to server...
Received from server: {"msgType": "Connected"}
Received from server: {"msgType": "DataReceived"}
```

*Diagram from tsncd,*

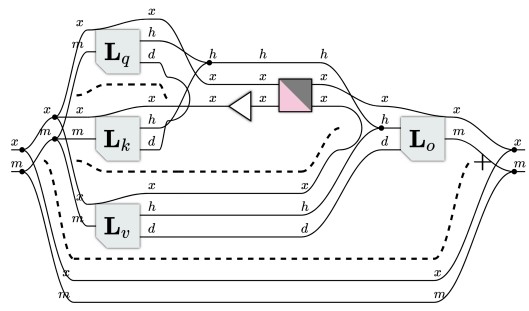

## Applying the configuration;

```
print('Scanning for Free Numerics,')
config = gc.NumericConfig.template(attention_block)
print(dpc.display_config(config))

print('\nApplying Configuration,')
ATTN_INNER, ATTN_HEAD, MODEL_DIM = (128, 8, 1024)
config.assign_values(d=ATTN_INNER, h=ATTN_HEAD,
m=MODEL_DIM)
print(dpc.display_config(config))
configured_attention =
config.apply_context(attention_block)

await print_term(configured_attention)
```

```
Scanning for Free Numerics,
Name|Type       |Bucket  |Assignment
x    |FreeNumeric|        |
m    |FreeNumeric|        |
h    |FreeNumeric|        |
d    |FreeNumeric|        |
d    |FreeNumeric|        |

Applying Configuration,
Name|Type       |Bucket  |Assignment
x    |FreeNumeric|        |
m    |FreeNumeric|2       |Integer(_value=1024)
h    |FreeNumeric|1       |Integer(_value=8)
d    |FreeNumeric|0       |Integer(_value=128)
d    |FreeNumeric|0       |Integer(_value=128)
Sending term to server...
Received from server: {"msgType": "Connected"}
Received from server: {"msgType": "DataReceived"}
```

*Diagram from tsncd,*

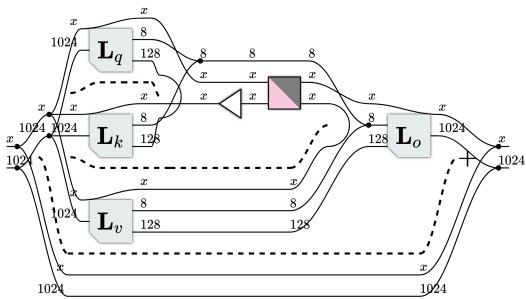

## PyTorch compilation

```python
import torch
import torch_compile.torch_compile as tc
import einops
from torch_compile.torch_utilities import
Multilinear

## An implemented torch module
def weighted_triangular_lower(x: torch.Tensor) ->
torch.Tensor:
    trilled = torch.tril(x)
    return trilled / (torch.sum(trilled, dim=-1,
keepdim=True) + 1e-8)

class MultiHeadAttention(torch.nn.Module):
    def __init__(self, d: int, h: int, m: int):
        super().__init__()
        self.d, self.h, self.m = d, h, m
        self.Lq, self.Lk, self.Lv = [Multilinear(m,
(h, d), bias=False) for _ in range(3)]
        self.Lo = Multilinear((h, d), m, bias=False)

    def forward(self, x):
        q, k, v = (self.Lq(x), self.Lk(x),
self.Lv(x))
        qk = einops.einsum(q, k, '... q h d, ... x h
d -> ... h q x')
        qk = torch.softmax(qk, dim=-1)
        qk = weighted_triangular_lower(qk)
        qkv = einops.einsum(qk, v, '... h q x, ... x
h d -> ... q h d')
        return self.Lo(qkv) + x

module_nn = MultiHeadAttention(ATTN_INNER,
ATTN_HEAD, MODEL_DIM)
module_cat =
tc.ConstructedModule.construct(configured_attention)

with torch.inference_mode():
    for p0, p1 in zip(module_nn.parameters(),
module_cat.parameters()):
        p0.data = p1.data.clone()

    BATCH_SIZE, SEQ_LEN = 8, 16
    dummy_data = torch.randn(BATCH_SIZE, SEQ_LEN,
MODEL_DIM)

    y_cat = module_cat(dummy_data)[0]
    y_nn = module_nn(dummy_data)
    print(torch.equal(y_cat, y_nn))
```

```
True
```

## Graph Processing

```python
import graphs.Hypergraph as hg
import graphs.Hypergraph2Morphism as h2m
graph =
hg.StructuredHypergraph.from_morphism(attention_bloc
k)
reconstructed = h2m.hypergraph_to_morphism(graph)
await print_term(reconstructed)
```

```
Sending term to server...
Received from server: {"msgType": "Connected"}
Received from server: {"msgType": "DataReceived"}
```

*Diagram from tsncd,*

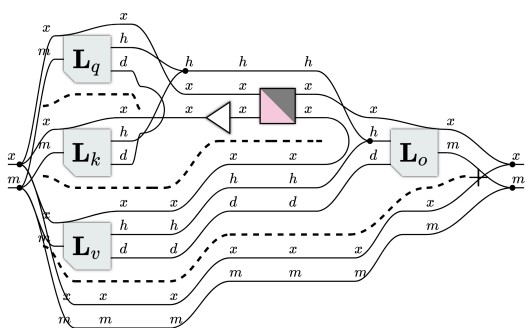

# DeepSeek-V3 and Mixture-of-Experts Code Generation

```python
import construction_helpers as ch # Needed for @
auto-alignment
import data_structure.Category as cat
import data_structure.Numeric as nm
import data_structure.Operators as ops
import data_structure.Term as fd
import display as dpl
import deepseek.data_structure as ds
import asyncio
import websocket_transfer.websockets_transfer as wst
import graphs.Hypergraph2Morphism as h2m
```

```python
## Helper functions to set up descriptive blocks and
blocks that expand as we hover.
## h2m.recylce uses the graph code to reconstruct an
expression in a standard format.
def standard_block(target: cat.BroadcastedCategory,
color: str, short_name: str | None = None, title:
str | None = None):
    title = title or short_name
    block = cat.Block(
            h2m.recycle(target),
            block_tag=cat.BlockTag(
                aesthetics=cat.BlockAesthetics(
                    title=title,
                    fill_color=color)))
    return block
def hover_block(target: cat.BroadcastedCategory,
color: str, short_name: str | None = None, title:
str | None = None):
    block = standard_block(target, color,
short_name, title)
    block_operation =
ops.BlockOperator.template(block, short_name)
    return block_operation
```

## Mixture of Expert Generation - with Sparse

```python
# The MoE on its own
gate = standard_block(
    ops.Linear.template('m', 1, 'L^g') @
ds.TopK.template() @ ops.SoftMax.template(),
    '#DFF7C3', 'Gate'
)

multihead_perceptron = standard_block(
    (0, 0) @
    ((ops.Linear.template(1, 2, 'W_0') @
ops.Elementwise.template())
     * ops.Linear.template(1, 2, 'W_1')) @
    ops.Einops.template('n f, n f -> n f') @
    ops.Linear.template(1, 2, 'W_2') @
    ops.Einops.template('k k m -> k m'),
    '#EDFCFF'
)

mixture_of_experts = (
    (0, 0) @
    (gate * multihead_perceptron) @
ops.Einops.template('n, n m -> m')
)
## Recycling reconstructs an expression from a block
await wst.send_term(h2m.recycle(mixture_of_experts))
```

## Mixture of Expert Generation - Complete Form

```python
k_axis = fd.DynamicName('k').capture(cat.RawAxis())
m_axis = fd.DynamicName('m').capture(cat.RawAxis())
f_axis = fd.DynamicName('f').capture(cat.RawAxis())
n_numeric =
fd.DynamicName('n').capture(nm.FreeNumeric())
n_count = cat.Natural(n_numeric)
n_axis =
fd.DynamicName('n').capture(cat.RawAxis(_size=n_nume
ric))
input_weaves = (
    cat.Weave(n_count, (cat.WeaveMode.TILED,)),
    cat.Weave(cat.Reals(), (m_axis,)),
)
output_weaves = (
    cat.Weave(cat.Reals(), (cat.WeaveMode.TILED,
f_axis)),
)
reindexings = (
    cat.Rearrangement((0,), (k_axis,)),
    cat.Rearrangement((), (k_axis,)),
)
W_in = cat.Broadcasted(
    operator=ops.Linear(fd.DynamicName('W',
fd.DynamicName('\\text{in}'))),
    input_weaves=input_weaves,
    output_weaves=output_weaves,
    reindexings=reindexings
)
kn_array = W_in.dom()[0]
m_array = W_in.dom()[1]

W_out = cat.Broadcasted(
    operator=ops.Linear(fd.DynamicName('W',
fd.DynamicName('\\text{out}'))),
    input_weaves=(
        cat.Weave(n_count, ()),
        cat.Weave(cat.Reals(), (f_axis,))
    ),
    output_weaves=(
        cat.Weave(cat.Reals(), (m_axis,)),
    ),
    reindexings=(
        cat.Rearrangement((), ()),
        cat.Rearrangement((), ())
    )
)

expert = (
    cat.Rearrangement(
        (0, 0, 1, 0, 1),
        (kn_array, m_array)
    ) @
    (kn_array * (((W_in @
ops.Elementwise.template()) * W_in) @
ops.Einops.template(',->'))) @
    W_out)

expert_gate = ops.Linear.template(1, n_axis) @
cat.Broadcasted(
    operator=ds.TopK(),
    input_weaves=(
        cat.Weave(cat.Reals(), (n_axis,)),
    ),
    output_weaves=(
        cat.Weave(cat.Reals(), (k_axis,)),
        cat.Weave(n_count, (k_axis,))
    ),
    reindexings=(
        cat.Rearrangement((), ()),
    )
) @ ops.SoftMax.template(contracted=True)

send_term = (
```

```
    (0, 0) @
    (expert_gate * cat.Array(cat.Reals(),
(m_axis,))) @
    (cat.Array(cat.Reals(), (k_axis,)) * expert) @
    ops.Einops.template('k, k m -> m')
)

await wst.send_term(h2m.recycle(send_term))
```

## DeepSeek-V3 - Full Generation

```
## SETUP MULTIHEAD LATENT ATTENTION with HOVER
BLOCKS
rotary_linear = hover_block(
    (ds.ComplexRotary().template()
     * (cat.RawAxis.named('x') >>
ops.Linear.template(1, 2,
output_datatype=ds.Complex()))) @
    ops.Einops.template('x d/2, x h d/2 -> x h d/2',
datatype=ds.Complex()) @
    ds.Decomplex.template(),
    '#FFE4D2', 'Ro', title='Rotary Linear'
)

qk_matmul = ops.Einops.template('q h d, x h d -> h q
x')

complex_s = (
    (0, 0) @
    (rotary_linear * rotary_linear) @
    qk_matmul
)

real_qkv = hover_block(
    (0, 0) @
      (ops.Linear.template(1, 2, 'Q')
    * (ops.Linear.template(1, '\\ell', 'L') @
      ops.Normalize.template() @
      (0, 0) @
      (ops.Linear.template(1, 2, 'K')
        * ops.Linear.template(1, 2, 'V')))),
    '#EFE1FF', 'QKV'
)

multihead_latent_attention = hover_block(
    (0, 0) @
    (complex_s * (real_qkv @ qk_matmul)) @
    ops.AdditionOp.template() @
    ops.SoftMax.template() @
    ops.WeightedTriangularLower.template() @
    ops.Einops.template('h q x, x h d -> q h d') @
    ops.Linear.template(2, 1, 'O'),
    '#FFF4E2', 'At', 'Multihead Latent Attention'
)

def multilayer_perceptron(inner_axis_name: str):
    return hover_block(
        (0, 0) @
        ((ops.Linear.template(name='W_0') @
ops.Elementwise.template())
            * ops.Linear.template(name='W_1')) @
        ops.Einops.template('2f, 2f -> 2f') @
        ops.Linear.template(inner_axis_name,
name='W_2'),
        '#EDFCFF', f'M_{{{inner_axis_name}}}',
title=f'Multilayer Perceptron ({inner_axis_name})'
    )

gate = hover_block(
    ops.Linear.template(1, 1, 'L^g') @
ds.TopK.template() @ ops.SoftMax.template(),
    '#DFF7C3', 'Gate'
)
```

```
multihead_perceptron = hover_block(
    (0, 0) @
    ((ops.Linear.template(1, 2, 'W_0') @
ops.Elementwise.template())
        * ops.Linear.template(1, 2, 'W_1')) @
    ops.Einops.template('n f, n f -> n f') @
    ops.Linear.template(1, 2, 'W_2') @
    ops.Einops.template('k k m -> k m'),
    '#EDFCFF', 'MoE', 'Mixture of Experts'
)

mixture_of_experts = hover_block(
    (0, 0) @
    (((0, 0) @ (gate * multihead_perceptron) @
ops.Einops.template('n, n m -> m'))
        * multilayer_perceptron('2f')) @
    ops.AdditionOp.template(),
    "#DDBBFF", 'FF', 'Mixture of Experts with
Gating'
)

def resnet(target):
    return ((0, 0) @ ops.Normalize.template() @
target @ ops.AdditionOp.template())

deepseek_block = resnet(multihead_latent_attention)
@ resnet(mixture_of_experts)

deepseek_model = standard_block(
    ops.Embedding.template('V', 1) @
    resnet(multihead_latent_attention) @
    resnet(multilayer_perceptron('d*f')) @
    cat.Block(
        body=deepseek_block,
        block_tag=cat.BlockTag(
            repetition=nm.Integer(27),

aesthetics=cat.BlockAesthetics(fill_color='#FFFFFF')
        )
    ) @
    ops.Normalize.template() @
    ops.Linear.template(1, 'V'),
    '#FFFFFF', 'DeepSeek', 'DeepSeek-V3'
)

recycled_model = h2m.recycle(deepseek_model)
await wst.send_term(recycled_model)
```

```
## We can generate a config for DeepSeek-V3,
extracting the configurable degrees of freedom.
import term_utilities.generate_config as gc
import display.display_config as dpc

config = gc.NumericConfig.template(recycled_model)
print(dpc.display_config(config))
```

| Name | Type | Bucket | Assignment |
|------|------|--------|------------|
| V | FreeNumeric | | |
| m | FreeNumeric | | |
| x | FreeNumeric | | |
| \ell | FreeNumeric | | |
| h | FreeNumeric | | |
| d | FreeNumeric | | |
| d | FreeNumeric | | |
| d/2 | FreeNumeric | | |
| d*f | FreeNumeric | | |
| n | FreeNumeric | | |
| k | FreeNumeric | | |
| f | FreeNumeric | | |
| 2f | FreeNumeric | | |
| V | FreeNumeric | | |

