# OpenReview forum: "Weaves, Wires, and Morphisms: Formalizing and Implementing the Algebra of Deep Learning"
_TMLR — Under review for TMLR_

### Review · Reviewer_rJxD · 2026-05-11

**Summary Of Contributions:**

The paper attempts to build a formal mathematical language for deep learning models, including indexing, tensors, broadcasting, axis manipulation, and related operations.

**Strengths:**

1. The paper addresses a well-motivated research question: how to better represent deep learning models in a formal and systematic way.
2. The transformation of operations into weaves and wires may help improve the representation and interpretability of many common operations, such as SoftMax, attention, and convolution.

**Weaknesses, Questions, and Suggestions:**

1. It remains unclear whether transforming operations into weaves and wires can help us understand the internal latent representations learned by deep neural networks. The framework appears to mainly formalize architectural and tensor-level operations, but its connection to feature-level interpretability is not yet clear.
2. The results are mainly constructive. Are there empirical results demonstrating efficiency improvements or practical benefits from applying the proposed framework?
3. Deep learning is developing rapidly. It would be helpful to include more concrete examples involving larger-scale models, such as modern language models or diffusion models. Since the paper provides a PyTorch implementation, it seems plausible that the framework could be extended to these settings.
4. A more direct comparison with existing tensor representation systems would strengthen the paper. For example, the authors could compare the proposed framework against representations based on PyTorch, NumPy, named tensors, or other tensor-programming systems, and discuss what advantages the proposed weave-and-wire representation provides.

**Audience:**

No

**Audience Explanation:**

The overall idea is interesting, as the paper provides a different perspective on interpreting deep learning through formal mathematical language. However, some concepts in the paper may make the proposed representation harder to understand.

The paper describes model components as morphisms and tensor axes as wires. However, many researchers working on deep learning or formal languages may not necessarily have a background in circuit diagrams or categorical diagrammatic notation. As a result, this transformation may make the concepts less intuitive rather than easier to interpret for some readers. I also have a follow-up question: what do the authors aim to demonstrate through the TypeScript-based rendering of learning diagrams? Is the TypeScript implementation mainly intended for visualization, user interaction, or validating the portability of the representation?

As a follow-up to my previous comments, it would be helpful if the authors could show whether the proposed representation and PyTorch demonstration can scale to existing popular models, such as Transformer-based language models or diffusion models with the complete diffusion process. Including such examples would likely attract a broader audience and improve the significance of the paper.

**Broader Impact Concerns:**

No specific ethical flags. Authors can add a section discussing about broader impacts.

**Claims And Evidence:**

Yes

**Claims Explanation:**

Most of the paper’s claims are reasonable, but the current results are mainly constructive and lack concrete empirical verification. Since the authors demonstrate that several common deep learning operations can be represented and implemented in PyTorch, applying the proposed mathematical framework to larger-scale models, such as Transformers or language models, would make the work more convincing. In addition, the authors may consider comparing their framework with other tensor representation systems under the same implementation setting, such as PyTorch-based representations, to provide a fairer and more direct evaluation of the proposed approach.

**Requested Changes:**

1. Demonstrate whether the proposed framework can be applied to feature-level representations and help explain how latent representations in deep learning models interact.

2. Provide empirical comparisons with related work, such as demonstrating efficiency improvements through implementation-level evaluation.

3. Include empirical implementations showing that the proposed method can be applied to modern architectures, such as Transformers, language models, or diffusion models.

4. Provide a direct comparison with related representation systems under the same PyTorch-based setting, including at least one case study to demonstrate the superiority or practical advantages of the proposed method.

---

> ### Author Response · Authors · 2026-06-29
>
> ## Weaknesses, Questions, Strengths / Requested Changes
> 1. **Representing Latent Representations** The aim of this framework is to represent deep learning _architectures_ as a topic of study independent from latent representations. In the background we have clarified that "black-box" behaviour, pertaining to the patterns in latent representations, is an emergent property from architectures, which is the focus of this work. It may be possible to use the formal representations to address latent representations in a procedural manner, though that is not the focus of this work.
> 2. **Constructive Nature of Results (shared)**
> 3. **Capturing Large Models (shared)**
> 4. **A Direct Comparison to Existing Representations** This is addressed in prior TMLR work, [Neural Circuit Diagrams](https://openreview.net/forum?id=RyZB4qXEgt), which this paper seeks to formalize. The same shortfalls of existing representations - principally ad-hoc diagrams and linear algebra notation - applies.
>
> **On the purpose of the TypeScript rendering** The TypeScript implementation serves two purposes: (i) **visualization** — it houses the diagram-rendering engine; and (ii) **validating portability of the representation** — the *same* constructed terms are serialized to JSON and exchanged between the Python package (which provides the algebraic interface and manipulation tools) and the TypeScript package (which provides the rendering engine) over WebSockets. That one representation moves losslessly between two independent language implementations is itself evidence that the representation is genuinely machine-portable, rather than an artifact of one toolchain. It is not intended as user-interaction validation.
>
>
> ## Relevance of Paper (given the Mathematical Complexity)
>
> As for relevance to the TMLR audience, the point about understadability for deep learning practitioners is noteworthy. We think there are two key aspects to this point. Firstly, the weaving approach does not _require_ the underlying category theory to be understood and, in practice, this approach does not introduce more implicit algebra than that which is already used. Secondly, this work serves as a foundational paper for future analysis of architectures using this formal approach and therefore requires the underlying category theory to be carefully laid out.
>
> Deep learning practitioners define broadcasting using a variety of tools which use complex implicit algebra. In PyTorch, broadcasting is defined in a manner akin to weaving through a combination of [Einops operations](https://github.com/arogozhnikov/einops), PyTorch broadcasting semantics (both [explicit](https://docs.pytorch.org/docs/2.12/generated/torch.vmap.html) and [implicit](https://docs.pytorch.org/docs/2.12/notes/broadcasting.html)). These implement similar techniques to weaving, indicating how output axes relate to inputs. The weaving approach, in contrast, provides a consistent representation and formal basis which relates to diagrams. By interfacing with algebraic code in Python which maps to diagrams in a similar manner to preexisting tools, the weaving approach can be used without the underlying category theory needing to be internalized.
>
> The depth of this paper is a consequence of being a foundational work. It supplies the confidence needed for the broadcasting and diagramming tools to be well-defined, and opens up further formal analysis of deep learning architectures. Prior Neural Circuit Diagram work such as [FlashAttention on a Napkin](https://openreview.net/forum?id=pF2ukh7HxA) provide useful analytical tools but, due to lacking a formal categorical description, lack an implementation. In this regard, category theory is unavoidable. Category theory is the mathematics of abstraction and composition. Attempting to describe composed deep learning architectures at multiple levels of abstraction will either use category theory or incidentally recreate many of its tools.
>
> Naturally, presenting a new mathematical framework does require managing familiarity with theoretical robustness. In this respect, we have modified the paper to more succinctly lay out the category theory. Major changes have been made to the sections introducing category theory and imposing the required structure for deep learning models in a manner which is both less dense and more general.

---

> > ### Comment · Reviewer_rJxD · 2026-07-11
> >
> > Thank you for the response. I have no further questions.

---

### Review · Reviewer_6Xvp · 2026-06-08

**Summary Of Contributions:**

This paper covers a framework for representing deep neural network architectures via algebra and category theory. The framework allows for reversible human and machine-interpretable representations. The authors spend most of the article defining the framework (a contribution) and then the remainder working through some implications and proof by examples of the framework's representation capacity (the second contribution).

**Additional Comments:**

I think this is an interesting framework and the visualizations are compelling. However, I have concerns with the complexity of even a simple deep neural network architecture in this representation. If human usability is a concern for the authors this complexity could be a hindrance.

I'd encourage the authors to consider taking the time to conduct a human subject study around the usability of their framework. I think this would provide the clearest signal of usability and would help it stand out compared to prior work.

**Audience:**

Yes

**Audience Explanation:**

There is a large body of work around AI theory and category theory as such I do not doubt that some of the individuals in the TMLR audience would find this work of interest.

**Broader Impact Concerns:**

The only related broader impact concerns come up with the speculation around AI acceleration in the final section. I don't feel this needs any further engagement given its speculative nature.

**Claims And Evidence:**

Yes

**Claims Explanation:**

For the most part the claims are what they are, since all results/"experiments" are in the paper by definition of them being consequences of the author's definitions (also in the paper). There are some claims later in the paper about this helping with AI assisted acceleration of science, but since these are speculative it's understandable that they are not fully supported.

The one case where I do have concerns around the support for the work is that the authors make a number of allusions to human interpretability and related concepts, "can be manipulated by both humans and machines". None of these associated claims are fully supported since the paper lacks a human subject study.

**Requested Changes:**

I have three major concerns with the paper in its current state.

My first concern is with the lack of a clear argument for value for the work. The authors focus a large portion of the paper on broadcasting, but they don't make any clear arguments about why the failure of representation of broadcasting matters. The authors also make a number of arguments related to human interpretability but never explicitly make the case that this framework will benefit that. The authors end with some speculation around AI assisted acceleration but this is speculation. The paper feels like it's missing a clear argument in the introduction as to why this work is valuable.

My second concern is with the lack of engagement with prior work. There is a great deal of prior work in the area [1], but most is not mentioned in the paper. This is especially concerning due to the lack of mention of prior category theory-based frameworks [2,3,4]. One of the ways the authors could alleviate my first concern is to engage with this prior work and identify their failures relative to their own framework. I'd like to see the authors include some kind of related work or background section.

My third concern is with the lack of results. The authors present a number of proofs by example as their results, but these essentially indicate that these kinds of representations are possible. It doesn't tell us anything about the utility of the representation. A comparison to prior work (i.e. "we can represent X but Y can't") would be one option here. But likely more useful would be a human subject study to confirm the usability/interpretability of the framework.

1. Youmbi, Serge. "A Survey of Category Theory and Deep Learning." (2025).

2. Chitukoori, Vinaykumar. "A Category Theory-Based Framework for Deep Neural Networks: Enhancing Structure and Interpretation." Proceedings of the Intelligent Robotics FAIR 2025. 2025. 64-68.

3. Chen, Yiting, Zhanpeng Zhou, and Junchi Yan. "Going beyond neural network feature similarity: The network feature complexity and its interpretation using category theory." International Conference on Learning Representations. Vol. 2024. 2024.

4. Healy, Michael John, and Thomas Preston Caudell. "Neural networks, knowledge and cognition: A mathematical semantic model based upon category theory." (2004).

---

> ### Author Response · Authors · 2026-06-29
>
> ## Requested Changes
> 1. **Clear Argument for Value** We have modified the background to better emphasise the need for formalized broadcasting. We emphasise the relationship between deep learning architectures and parallelized code, which implies that a formal description requires a manageable definition of parallelized, broadcasted operations. The subsequent contributions of the work, then, can be seen as scaffolding this definition and following through on the natural implications, including developing a codebase which is able to algebraically manipulate and represent deep learning architectures.
> 2. **Engagement with Prior Work** In the background, we have included an overview of prior category theory-deep learning work. A direct comparison with this work is not fully relevant. Much of the category theory-deep learning work uses a basic categorical scaffold of deep learning architectures to reframe an analysis using the tools of category theory. In contrast, we focus on enhancing this scaffold by equipping it with formal broadcasting, therefore allowing architectures to be dealt with as they appear in practice. Our work therefore enhances, rather than competes with, this prior work.
> 3. **Lack of Results (see shared themes)**

---

### Review · Reviewer_CDBG · 2026-06-19

**Summary Of Contributions:**

This paper proposes a categorical framework for formalizing the algebraic calculation structure in neural network architectures, and provides interface for visualizing them. The main motivation is that existing descriptions of neural tensor operations often leave transformations implicit and overly coarse. Within the defined framework, the paper introduces a construction system that connects symbolic entities and tensor computation types in the DL environment. Beyond formally defining them, the paper also introduces visualization tools to represent how operations are transformed over tensor axes. The framework is illustrated on softmax, Einstein summation, Transformer block, etc, and the diagram rendering shows reasonable results.

*Key Strengths*: The tensor-level calculations and structural dynamics that are usually implicit in code. This proposed categorical framework and implemented visualization tools would be useful to model analysis and development.

*Key Weaknesses*: Some mathematical definitions are not yet supported by rigorous evidence, especially with regard to the closure property. In addition, the applicability to modern deep learning systems such as routing, mixed precision, and large-scale compilation remains insufficiently clarified, making the claims about deep learning models somewhat overstated.

**Additional Comments:**

This is the first time I have reviewed a paper of this type, and honestly, I was initially unsure whether it fell within the scope of TMLR. I therefore spent some time looking into this and found that similar papers had previously been accepted. Based on this, I do not raise the  scope of this paper as a concern.

**Audience:**

Yes

**Audience Explanation:**

1. The calculation details of neural networks are often implicit and entangled with implementation details in code. A formal framework that makes these structures explicit could therefore be valuable for model analysis and understanding.
2. The authors do not merely propose a philosophical or abstract categorical framework. They also attempt to connect the framework to executable PyTorch modules. This connection to implementation makes the work more relevant to readers interested in practical deep learning systems.
3. The proposed framework has the potential to support future neural network developers in reasoning about tensor data structures and algebraic operations at multiple levels of abstraction. This could make the design and analysis of neural network components more transparent, as suggested by the forward-looking discussion in Figure 24.

**Broader Impact Concerns:**

N/A. This paper does not have ethical implications.

**Claims And Evidence:**

No

**Claims Explanation:**

1. The authors should provide a clear proof that finite affine transformations are closed over finite axes. The constraint that the "image is captured by the codomain" also needs to be handled more carefully, especially with respect to whether and how this property is preserved.
2. Some stochastic operations that commonly appear in modern deep learning systems are not discussed in detail. However, its applicability to operations such as dropout, mixed-precision computation, and MoE routing remains unclear. The proposed framework appears well suited to deterministic tensor programs with static shapes and clean broadcasting semantics, but it remains unclear how it can be grounded in the dynamic states described above. For example, the Axis–Stride Category $\mathsf{St}$ seems to only work well on compile-time static shape information. Without further clarification, the broad claims about deep learning models are somewhat overstated.
3. In the real-world application, the users would expect a mechanism for ensuring the safety of the proposed transformations. The authors could provide a compiler correctness theorem or other well-defined guarantees showing that tensor computation graphs are transformed in a fault-tolerant manner.
4. Although the authors state that the paper is constructive rather than benchmark-driven, it would still be helpful to include an estimate of the compilation budget required by the proposed framework. In particular, the paper should clarify what scale of neural network, in terms of parameter count or graph size, can be supported within a practically acceptable compilation time.

**Requested Changes:**

1. The authors may strengthen the formal definitions of St. Please explicitly define the closure properties of Finite affine transformations.

2. Please clarify the scope of the framework with respect to operations such as dropout, mixed-precision computation, and MoE routing.

3. Please strengthen the implementation evaluation. The authors could include systematic tests for alignment.

4. Please include at least a rough estimate of the compilation budget and scalability of the proposed approach, for example in terms of graph size, model parameter count, or acceptable compilation time.

5. Please formalize the autoalignment mechanism in the result section.

6. The paper introduces many terms and morphisms. A compact glossary would improve readability (the authors may expand the scope of Table 1).

7. Appendix section numbering appears confusing, with subsections labeled as 1.1.1, 1.1.2, etc. inside Appendix B. This should be cleaned up, e.g., B.1.1.1.

---

> ### Author Response · Authors · 2026-06-29
>
> The section introducing the categorical framework and extending it to deep learning models has been rewritten to address these comments. Instead of defining broadcasting just for deep learning models, we have created a generic categorical construct $[\mathcal{C}, \mathcal{I}]$ which lifts a base category $\mathcal{C}$ using the stride operations supplied by $\mathcal{I}$. This addresses the key points:
>
> 1. **Affine Transforms** Composition within the category of finite affine transformations is fully laid out, showing how the linear and offset components compose.
> 2. **Stochastic Operations and Static/Dynamic Sizes** By drawing from $\mathbf{BorelStoch}$, randomness is integrated into the category representing deep learning architectures. Throughout the paper, we have also clarified how UIDs can be used to separate dynamic (unassigned) and static (assigned) sizes.
> 3. **Safety of Transformations** In Section 4.2, we have described how a category of algorithms $\mathbf{Alg}$ can be used to describe multiple computational schemes of the same mathematical operation. Categorical transformations, such as transformations to and from hypergraphs or bifunctorial rearrangements, are mathematically robust, drawing from an extensive catalogue of prior work.
> 4. **Compilation Time** The compilation budget of the framework is negligable. A complex model such as _DeepSeek-V3_ can be represented with ~30 components. Components generally interact only with local components directly before and after, and hypegraph analysis uses a cache to store connections. Therefore, the memory and time required to analyze an expression scales linearly with the small number of components in use.
>
> ## Requested Changes
> 1. The formal definition of the underlying category and its parallelization have been greatly strengthened.
> 2. Dropout and MoE routing have been elaborated in the key operations section. Mixed-precision computation is a computational aspect outside of the scope of this work, though a comment has been made in the last paragraph of Section 4.2.
> 3. In 1.1.3 of the attached code, the compiled PyTorch code is shown to be equivalent to explicit PyTorch code. This code attached to this work aims to be a proof-of-concept, laying the foundations for a fully-developed codebase.
> 4. Computational time is negligable. It likely scales linearly with component count, though all analyses in this work are performed in well under 1s.
> 5. The autoalignment section has been elaborated as the application of functors as needed.
> 6. A compact glossary has not been included, though can be if still desired. The rewriting of the categorical section should address many of the clarity concerns.
> 7. The Appendix has been reduced and cleaned up.

---

### Author Response · Authors · 2026-06-29
**Response to Shared Themes**

To the reviewers, thank you for taking your time to assess this work. As this work aims to define a new area and contains a number of novel concepts, we greatly appreciate your feedback. Your comments have been helpful in tightening and more clearly expressing our work.

Following the reviewer comments, we have made major changes to the work, including:
- Rewriting the background to better emphasise broadcasting,
- Rewriting the introduction of the categorical structure,
- Integrating a formulation of DeepSeek-V3 into the results.

In the reviewer comments, there were a number of shared themes which we will address collectively.

## Constructive Nature of Results
A major point of contention was the constructive nature of the results. TMLR's submission guidelines focuses on claims supported by evidence and relevance. The lack of a formalized definition of deep learning architectures, in a field as mathematically sophisticated as AI, is a clear gap which can provide value if addressed. It would allow how we approach deep learning models to be proceduralized. For instance, the lack of a formalized definition precludes compilation from mathematical formulae to efficient, parallelized low-level code. By providing mathematical definitions and a systematic code implementation that goes all the way to representing the components of DeepSeek-V3, this paper addresses this challenge. Though this paper lacks the empirical tables and graphs typically associated with a deep learning paper, it fits within TMLR's submission guidelines, including:
> development of new analytical frameworks that advance theoretical studies of practical learning methods;

> new approaches for analysis, visualization, and understanding of artificial or biological learning systems;

Furthermore, a paper which introduces this mathematics and codebase _as well as_ an empirical result (e.g. showing that compilation to low-level code works and empirically matches an estimated performance model, or error propagation matches estimations) would include the development of the mathematics and codebase as a largely independent contribution. Large parts of that mathematics and codebase may be used for other investigations. Indeed, such an investigation would be more mathematically robust if its specific analysis (e.g. low-level kernels) is an extension of a universal description of deep learning architectures, rather than an ad-hoc, bespoke framework. Therefore, it makes sense for the mathematics and codebase to be provided in a separate, foundational work. Given the submission guidelines of TMLR, it is an appropriate venue for such a work to be presented.

## Capturing Large Models
The paper has been modified to culminate in presenting the entirety of _DeepSeek-V3_, diagramming MoEs, complex rotaries, and other operations. We will supply PyTorch code to allow the compilation of a small, toy model in the coming days. For now, the paper has been updated to describe the necessary details.